# Towards Doctor-Like Reasoning: Medical RAG Fusing Knowledge with Patient Analogy through Textual Gradients

**Yuxing Lu**[1,2†]**, Gecheng Fu**[3†]**, Wei Wu**[1]**, Xukai Zhao**[4]**, Goi Sin Yee**[5]**, Jinzhuo Wang**[1‡]

[1] Department of Big Data and Biomedical AI, Peking University, Beijing, China
[2] Wallace H. Coulter Department of Biomedical Engineering, Georgia Institute of Technology, Atlanta, USA
[2] Department of Biomedical Engineering, Peking University, Beijing, China
[4] School of Architecture, Tsinghua University, Beijing, China
[5] School of Life Sciences, Peking University, Beijing, China

† Equal contribution    ‡ Corresponding author: `wangjinzhuo@pku.edu.cn`

## Abstract

Existing medical RAG systems mainly leverage knowledge from medical knowledge bases, neglecting the crucial role of experiential knowledge derived from similar patient cases – a key component of human clinical reasoning. To bridge this gap, we propose **DoctorRAG**, a RAG framework that emulates doctor-like reasoning by integrating both explicit clinical knowledge and implicit case-based experience. DoctorRAG enhances retrieval precision by first allocating conceptual tags for queries and knowledge sources, together with a hybrid retrieval mechanism from both relevant knowledge and patient. In addition, a **Med-TextGrad** module using multi-agent textual gradients is integrated to ensure that the final output adheres to the retrieved knowledge and patient query. Comprehensive experiments on multilingual, multitask datasets demonstrate that DoctorRAG significantly outperforms strong baseline RAG models and gains improvements from iterative refinements. Our approach generates more accurate, relevant, and comprehensive responses, taking a step towards more doctor-like medical reasoning systems.

## 1 Introduction

Large Language Models (LLMs) hold considerable promise for transforming healthcare, offering capabilities to assist medical professionals, enhance patient care, and accelerate biomedical research [11]. Within this landscape, Retrieval-Augmented Generation (RAG) has emerged as a pivotal technique, grounding LLM outputs in factual knowledge to mitigate hallucinations and bolster reliability – qualities paramount in the high-stakes medical domain [30]. Existing medical RAG systems commonly incorporate extensive external knowledge bases containing clinical guidelines, evidence-based medicine, and research literature [46, 41, 42]. This approach, while valuable, overlooks a critical dimension of expert medical reasoning: the application of experience gained from encountering numerous patient cases over time.

Human clinicians routinely integrate formal medical knowledge (***Expertise***) with experiential insights gleaned from similar past situations (***Experience***, i.e., case-based reasoning) iteratively to make diagnoses, formulate treatment plans, and answer patient queries [10]. Existing medical RAG frameworks fail to capture the information from similar patients, limiting their ability to truly emulate doctor-like reasoning (Figure 1). This deficiency can hinder their overall performance in complex, context-dependent tasks such as differential diagnosis or the formulation of personalized treatment recommendations. Furthermore, standard retrieval methods (e.g., dense vector search) often struggle to capture the fine-grained semantic distinctions crucial for medical relevance [33]. Compounding this, the outputs of generative models require rigorous validation to ensure they are not only relevant

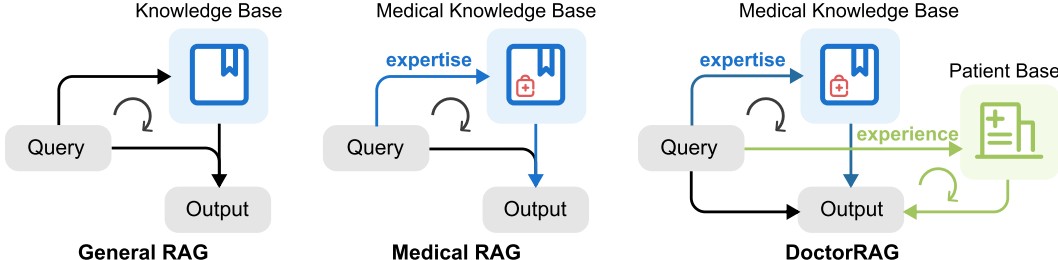

Figure 1: General and Medical RAGs retrieve solely from knowledge bases, whereas DoctorRAG enhance responses by integrating both knowledge-based **expertise** and case-based **experience**, mirroring clinical practice.

but also faithfully aligned with the retrieved medical context, a challenge that necessitates a dedicated mechanism for iterative refinement and grounding [18].

To address these limitations, we propose DoctorRAG, a novel RAG framework designed to bridge the gap between current systems and human clinical practices. The core innovation of DoctorRAG lies in its ability to emulate doctor-like reasoning by integrating both explicit clinical knowledge from KBs with implicit experiential knowledge derived from analogous patient cases. To effectively retrieve proper information from these heterogeneous sources, DoctorRAG first decomposes complex queries and knowledge chunks into declarative statements and conceptual tags. This structured representation enables a more precise hybrid retrieval mechanism that combines semantic vector similarity with targeted conceptual matching. The generated answer is then refined and validated by Med-TextGrad, a multi-agent textual gradient process that iteratively ensures the output adheres to the retrieved context and patient query, effectively grounding the response and ensuring the system's accuracy and reliability. Comprehensive experiments conducted on diverse multilingual and multitask datasets—encompassing disease diagnosis, question answering, treatment recommendation, and text generation demonstrate that DoctorRAG significantly outperforms strong RAG baselines in terms of accuracy, relevance, and faithfulness. Pairwise comparison between answers of different iteration further demonstrate Med-TextGrad paves the way for medical AI systems capable of more comprehensive, reliable, and clinically-informed reasoning.

## 2 Methodology

We propose the DoctorRAG framework for clinical decision support, designed to integrate domain-specific medical knowledge with patient-specific clinical data, and to refine the generated response through a multi-agent, iterative optimization process. This framework, illustrated in Figure 2, leverages concept matching and vectorized embeddings for retrieval, and a Med-TextGrad optimization strategy to iteratively refine generated answers through back propagation. The pipeline consists of two primary stages: (1) Expertise-Experience retrieval and aggregation, and (2) Iterative answer optimization using multi-agent textual gradients.

### 2.1 Expertise-Experience retrieval and aggregation

The retrieval and aggregation module (prompts in D.1, examples in E.1) operates on two primary data sources: a **Knowledge Base** ($\mathcal{K}$) and a **Patient Base** ($\mathcal{P}$).

The **Knowledge Base** is constructed from several medical knowledge sources. These sources are first segmented into textual chunks. Each chunk is then transformed into a declarative sentence (details in B.3), denoted $d_i$, using a Declarative Statement Agent $\text{Dec}_\phi$. Subsequently, each sentence $d_i$ is annotated with a set of medical concept identifiers $c_i = \text{Tag}_\phi(d_i)$, where $c_i \subseteq \mathcal{C}$ (details in B.4). The function $\text{Tag}_\phi$ represents a Query Tagging Agent, and $\mathcal{C}$ is a predefined controlled vocabulary of medical concepts (e.g., ICD-10 codes). The Knowledge Base is thus defined as the set of pairs $\mathcal{K} = \{(d_i, c_i) \mid i = 1, \ldots, N_\mathcal{K}\}$, where $N_\mathcal{K}$ is the total number of processed declarative sentences. Each sentence $d_i$ is encoded into a $D$-dimensional dense vector representation $v_{d_i} = \text{E}_{\text{emb}}(d_i) \in \mathbb{R}^D$. Here, $\text{E}_{\text{emb}}$ denotes the embedding function of an LLM.

The **Patient Base** comprises de-identified patient records, formally defined as $\mathcal{P} = \{(p_j, s_j, a_j) \mid j = 1, \ldots, N_\mathcal{P}\}$, where $N_\mathcal{P}$ is the count of patient records. For each record $j$, $p_j$ represents the patient's chief complaint or clinical conversation, $s_j$ denotes structured clinical data (e.g., diagnoses,

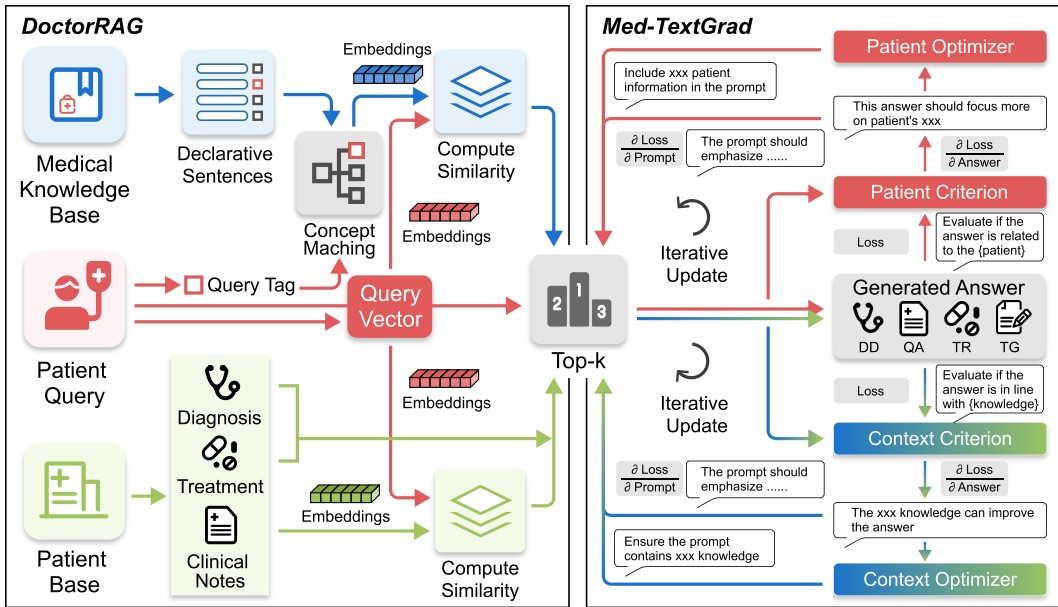

Figure 2: **Overview of the proposed DoctorRAG & Med-TextGrad framework.** Medical knowledge is transformed into declarative statements and tagged to match the patient's query. Dual-database retrieval generates a response incorporating both clinical expertise and experience. A two-way multi-agent Med-TextGrad optimization process refines the answer through an iterative computation graph involving textual gradient backpropagation.

treatments), and $a_j$ encompasses auxiliary metadata (e.g., demographics). The primary textual component $p_j$ is encoded into a vector $v_{p_j} = \mathrm{E}_{\mathrm{emb}}(p_j) \in \mathbb{R}^D$ using the identical embedding function $\mathrm{E}_{\mathrm{emb}}$. The remaining data $s_j$ and $a_j$ are preserved for contextual augmentation.

Given a user query $q$, it is processed through two concurrent pathways. Firstly, the Query Tagging Agent $\mathrm{Tag}_\phi$, annotates the query with a set of relevant medical concept identifiers $c_q = \mathrm{Tag}_\phi(q)$, where $c_q \subseteq \mathcal{C}$. Secondly, the query $q$ is embedded into the same $D$-dimensional latent space, yielding $v_q = \mathrm{E}_{\mathrm{emb}}(q) \in \mathbb{R}^D$. Retrieval from both databases proceeds as follows:

**Knowledge Retrieval:** To retrieve relevant declarative sentences from $\mathcal{K}$, a concept-constrained semantic similarity score $s_\mathcal{K}(q, d_i)$ is computed between the query $q$ and each sentence $d_i$ (associated with concepts $c_i$):

$$s_\mathcal{K}(q, d_i) = \begin{cases} \frac{v_q^\top v_{d_i}}{\|v_q\|\|v_{d_i}\|} & \text{if } c_q \cap c_i \neq \emptyset, \\ -\infty & \text{otherwise,} \end{cases} \tag{1}$$

where the fraction denotes the cosine similarity. This formulation prioritizes semantic relevance for sentences sharing common concept identifiers with the query. The $k$ knowledge entries $(d_i, c_i)$ achieving the highest non-negative similarity scores constitute the set $\mathcal{K}_{\text{top-}k}$.

**Patient Retrieval:** Concurrently, to retrieve relevant patient records from $\mathcal{P}$, the cosine similarity $s_\mathcal{P}(q, p_j)$ between $v_q$ and the embedding of each patient's textual data $v_{p_j}$ is computed:

$$s_\mathcal{P}(q, p_j) = \frac{v_q^\top v_{p_j}}{\|v_q\|\|v_{p_j}\|}. \tag{2}$$

The $k$ patient records $(p_j, s_j, a_j)$ yielding the highest similarity scores constitute the set $\mathcal{P}_{\text{top-}k}$.

Finally, the retrieved information is aggregated to construct a unified context $C$ for the answer generation module. This context is formed by concatenating the textual content from the top-$k$ declarative sentences and formatted textual representations of the top-$k$ patient records:

$$C = \mathrm{Concat}\left( \bigcup_{(d_i, c_i) \in \mathcal{K}_{\text{top-}k}} d_i, \bigcup_{(p_j, s_j, a_j) \in \mathcal{P}_{\text{top-}k}} \mathrm{Format}(p_j, s_j, a_j) \right), \tag{3}$$

where $\mathrm{Concat}(\cdot, \cdot)$ joins all textual elements, and $\mathrm{Format}(\cdot)$ converts a patient record tuple into a structured textual representation. This context $C$ concatenates both general medical knowledge (*expertise*) and patient-derived clinical evidence (*experience*), serving as input for generation.

## 2.2 Iterative answer optimization with multi-agent Med-TextGrad

To generate and refine answers, we propose Med-TextGrad, a multi-agent optimization framework adapted for the medical domain (Algorithm in B.2, prompts in D.2, examples in E.2). This framework leverages principles of TextGrad [43] for textual gradient-based optimization. The system architecture comprises a **Generator**, a **Context Criterion**, a **Patient Criterion**, and conceptually distinct **Context Optimizer** and **Patient Optimizer** roles guiding iterative refinement. The process uses a computation graph where textual gradients update the prompt and, consequently, the answer.

The fundamental goal of the Med-TextGrad framework is to identify an optimal prompt $P_{\mathrm{opt}}^{\star}$ that, given a query $q$ and its retrieved initial context $C$, generates the best possible answer $A_{\mathrm{opt}}^{\star} = \mathrm{Gen}_{\psi}(P_{\mathrm{opt}})$. 'Best' is defined in terms of minimizing a textual loss function $\mathcal{L}$, which incorporates critiques regarding medical context consistency and patient-specific relevance. This objective is analogous to finding the optimal parameters $\theta^*$ for a neural network by minimizing a loss function. Formally, the target optimal prompt $P_{\mathrm{opt}}$ is defined as:

$$P_{\mathrm{opt}}^{\star}(q, C) = \arg\min_{P} \mathcal{L}(A \mid q, C) \quad \text{s.t.} \quad P \in \mathcal{P}, \ A = \mathrm{Gen}_{\psi}(P) \tag{4}$$

where $\mathcal{P}$ represents the space of possible prompt space. The textual loss $\mathcal{L}(A \mid q, C)$ is associated with the answer generated from prompt $P$; it is explicitly defined in Equation 8 as the sum of critiques from different criteria. The Med-TextGrad iterative procedure, starting from an initial prompt and applying textual gradient descent steps (Equation 14), is designed to find a refined prompt $P_T$ that serves as an approximation to $P_{\mathrm{opt}}^{\star}$.

### 2.2.1 Computation graph with gradient backpropagation

The answer generation and iterative refinement process is modeled as a directed computation graph. The forward pass generates an answer from a prompt and evaluates it against specified criteria, yielding textual critiques that serve as a proxy for loss. The backward pass computes and propagates textual gradients from these critiques to guide prompt refinement. This is conceptualized as:

$$\text{Prompt} \xrightarrow{\text{Generator}} \text{Answer} \xrightarrow{\text{Criterion}} \text{Critiques (Loss Proxy)} \tag{5}$$

$$\text{Prompt} \xleftarrow{\nabla_{\text{Prompt}}} \text{Answer} \xleftarrow{\nabla_{\text{Answer}}} \text{Critiques (Loss Proxy)} \tag{6}$$

Here, $\nabla_{\mathrm{Answer}}$ and $\nabla_{\mathrm{Prompt}}$ represent the textual gradient computation steps. The input prompt $P_t$ at iteration $t$ is formulated as the concatenation of the initial query $q$ and the aggregated context $C$. An answer $A_t$ is produced by the **Generator**, and is assessed by the **Context Criterion** and **Patient Criterion** to produce textual critiques. The **Generator**, denoted $\mathrm{Gen}_{\psi}$, generates the answer $A_t$ based on the current prompt $P_t$:

$$A_t = \mathrm{Gen}_{\psi}(P_t) = \mathrm{Gen}_{\psi}(\mathrm{Concat}(q, C)). \tag{7}$$

The **Context Criterion** ($\mathrm{KC}_{\psi}$) and **Patient Criterion** ($\mathrm{PC}_{\psi}$) evaluate $A_t$. $\mathrm{KC}_{\psi}$ assesses factual alignment and consistency of $A_t$ with context $C$. $\mathrm{PC}_{\psi}$ evaluates relevance, appropriateness, and patient-specificity of $A_t$ to query $q$. The aggregated textual critique, representing the overall "loss", is given directly by $\mathcal{L}(A_t)$:

$$\mathcal{L}(A_t) = \mathcal{L}(A \mid q, C) = C_{\mathrm{KC}}(A_t) + C_{\mathrm{PC}}(A_t) = \mathrm{KC}_{\psi}(A_t, C) + \mathrm{PC}_{\psi}(A_t, q). \tag{8}$$

Ideally, this loss can also inform an early stopping strategy in Med-TextGrad.

### 2.2.2 Textual gradient computation and backpropagation

Gradients of the critiques with respect to the answer $A_t$ are computed independently. These gradients, $\frac{\partial \mathrm{KC}(A_t)}{\partial A_t}$ and $\frac{\partial \mathrm{PC}(A_t)}{\partial A_t}$, provide instructive feedback from each perspective. They are generated by a LLM $\mathcal{G}_{\mathrm{grad\_A}}$ prompted by $\psi_{\mathrm{grad\_A}}$ specific to each criterion:

$$\frac{\partial \mathrm{KC}(A_t)}{\partial A_t} = \mathcal{G}_{\mathrm{grad\_A}}(A_t, C, \mathrm{KC}(A_t); \psi_{\mathrm{grad\_A\_KC}}), \tag{9}$$

$$\frac{\partial \text{PC}(A_t)}{\partial A_t} = \mathcal{G}_{\text{grad\_A}}(A_t, q, \text{PC}(A_t); \psi_{\text{grad\_A\_PC}}). \tag{10}$$

Here, $\mathcal{G}_{\text{grad\_A}}$ takes the current answer, the relevant content (either context $C$ or patient query $q$), and the critique itself to produce a textual gradient aimed at improving the answer.

These answer-level textual gradients are natural language descriptions suggesting specific improvements for $A_t$ from the viewpoint of respective criterion. Subsequently, they are utilized to compute distinct textual gradients with respect to the prompt $P_t$. These prompt-directed gradients, $\frac{\partial \text{KC}(A_t)}{\partial P_t}$ and $\frac{\partial \text{PC}(A_t)}{\partial P_t}$, articulate how $P_t$ should be modified to elicit an improved $A_t$ that addresses the critiques from each specific criterion. These are generated by a LLM $\mathcal{G}_{\text{grad\_P}}$ prompted by $\psi_{\text{grad\_P}}$:

$$\frac{\partial \text{KC}(A_t)}{\partial P_t} = \mathcal{G}_{\text{grad\_P}}\left(P_t, A_t, \frac{\partial \text{KC}(A_t)}{\partial A_t}; \psi_{\text{grad\_P\_KC}}\right). \tag{11}$$

$$\frac{\partial \text{PC}(A_t)}{\partial P_t} = \mathcal{G}_{\text{grad\_P}}\left(P_t, A_t, \frac{\partial \text{PC}(A_t)}{\partial A_t}; \psi_{\text{grad\_P\_PC}}\right). \tag{12}$$

In these equations, $\mathcal{G}_{\text{grad\_P}}$ uses the current prompt, the generated answer, and the corresponding answer-level gradient to compute the prompt-level textual gradient.

The computed prompt gradients, $\frac{\partial \text{KC}(A_t)}{\partial P_t}$ and $\frac{\partial \text{PC}(A_t)}{\partial P_t}$, provide pathway-specific textual feedback for refining the prompt $P_t$. Conceptually, an overall textual gradient $\frac{\partial \mathcal{L}(A_t)}{\partial P_t}$ for the combined critique with respect to the prompt $P_t$ arises from the application of the chain rule. This can be expressed as:

$$\frac{\partial \mathcal{L}(A_t)}{\partial P_t} = \nabla_{\text{Gen}}\left(P_t, A_t, \text{Aggregate}\left(\frac{\partial \text{KC}(A_t)}{\partial A_t}, \frac{\partial \text{PC}(A_t)}{\partial A_t}\right)\right). \tag{13}$$

Here, $\text{Aggregate}(\cdot, \cdot)$ denotes the textual combination of the individual answer-level gradients, and $\nabla_{\text{Gen}}$ is the TextGrad operator that performs the backpropagation through the **Generator** $\text{Gen}_\psi$. This overall gradient illustrates how combined feedback theoretically propagates to the prompt.

### 2.2.3 Iterative optimization with context and patient optimizers

The iterative refinement of the prompt $P_t$ is guided by the distinct textual gradients computed in the previous step: the context-focused prompt gradient $\frac{\partial \text{KC}(A_t)}{\partial P_t}$ and the patient-focused prompt gradient $\frac{\partial \text{PC}(A_t)}{\partial P_t}$. The **Context Optimizer** and **Patient Optimizer** roles leverage these respective gradients to ensure the prompt evolves in a manner that addresses both alignment with medical context and relevance to the specific patient.

This is achieved through a Textual Gradient Descent (TGD) step, where the current prompt $P_t$ is updated. This update is performed by another LLM $\mathcal{G}_{\text{TGD}}$, which synthesizes the improvement suggestions from both prompt gradients to produce the refined prompt $P_{t+1}$:

$$P_{t+1} = \text{TGD.step}\left(P_t, \frac{\partial \mathcal{L}(A_t)}{\partial P_t}\right) = \text{TGD.step}\left(P_t, \frac{\partial \text{KC}(A_t)}{\partial P_t}, \frac{\partial \text{PC}(A_t)}{\partial P_t}\right). \tag{14}$$

The updated prompt $P_{t+1}$ is then fed back to the generator to produce a new answer $A_{t+1}$:

$$A_{t+1} = \text{Gen}_\psi(P_{t+1}). \tag{15}$$

This iterative cycle continues for a predetermined number of iterations $T$ (3 in our implementation), or until a suitable convergence criterion (e.g., minimal changes in critiques or the answer) is met. The final output of Med-TextGrad is the refined answer $A_T$.

## 3 Experimental Setup

### 3.1 Datasets

To comprehensively evaluate DoctorRAG, we utilized a diverse collection of datasets spanning multiple languages and covering key medical tasks: disease diagnosis (DD), question answering (QA), treatment recommendation (TR), and text generation (TG). Further details on each dataset are provided in Appendix B.1. An overview of the datasets, including their respective languages, medical fields, and sample counts for each task, is presented in Table 1.

|          | Dataset      | Domain     | DD        | QA  | TR     | TG     |
|----------|--------------|------------|-----------|-----|--------|--------|
| **Chinese** | DialMed [17]  | General    | 11,996    | –   | 11,996 | –      |
|          | RJUA [27]    | Urology    | 2,340     | –   | 2,340  | 2,340  |
|          | MuZhi [38]   | Pediatrics | 527       | –   | –      | –      |
| **English** | DDXPlus [14]  | General    | 1,292,579 | –   | –      | –      |
|          | NEJM-QA [5]  | Nephrology | –         | 655 | –      | –      |
|          | COD [6]      | General    | 39,149    | –   | –      | 39,149 |
| **French**  | DDXPlus [14]  | General    | 1,292,579 | –   | –      | –      |

Table 1: Datasets for evaluation. **DD**: Disease Diagnosis, **QA**: Question Answering, **TR**: Treatment Recommendation, **TG**: Text Generation. Numerical values indicate sample counts.

For Chinese-language tasks, we incorporated three datasets. DialMed [17], a general medicine dataset featuring structured dialogues, is employed for disease diagnosis and drug recommendation tasks. RJUA [27] focuses on urology and is used to assess DoctorRAG's capabilities in disease diagnosis, treatment recommendation, and the generation of informative responses. Muzhi [38] centered on pediatric care, comprises dialogues with symptom descriptions and diagnostic recommendations.

For English-language evaluations, we employed DDXPlus [14], which is a large-scale general medicine dataset that includes socio-demographic data, multi-format symptoms/antecedents, and differential diagnoses. The NEJM-QA [5], specializing in nephrology, provides 656 multiple-choice questions to evaluate question answering abilities. Additionally, COD [6], a general medicine dataset, is used to assess performance on disease diagnosis and medical text generation.

We included the French version of DDXPlus [14] to assess DoctorRAG's multilingual performance.

We constructed the external knowledge base for DoctorRAG using MedQA [20] datasets. MedQA encompasses bilingual Chinese-English medical question-answering pairs and medical textbooks, providing a foundation for cross-lingual knowledge retrieval. We also translated a subset of MedQA to French for the French version of DDXPlus [14]. Each knowledge entry is pre-segmented, transformed into declarative statements using the DeepSeek-V3 model [24], and annotated with corresponding concept labels to optimize DoctorRAG's medical knowledge retrieval capabilities.

For each dataset, we established a clear partitioning: approximately 80% of the patient records from each dataset were allocated for constructing DoctorRAG's patient base, while the remaining were held out as a distinct evaluation set. To ensure an unbiased assessment and prevent data leakage, we guarantee that for any given sample within this evaluation set, its corresponding full patient record and extreme similar ones (similarity > 0.99) were strictly excluded from DoctorRAG's patient base during evaluation of that specific sample. All samples are treated as independent clinical encounters.

### 3.2 LLM backbones

We employed a selection of advanced LLM backbones for comprehensive and balanced comparison. The primary aims were to assess the multilingual and multitask capabilities of these foundational models and to evaluate the robustness of DoctorRAG when integrated with different LLMs. The models selected include DeepSeek-V3[24], Qwen-3 [2], GLM-4-Plus [16], and GPT-4.1 [1]. To ensure consistency across all experimental procedures, DeepSeek-V3 was specifically assigned to data preprocessing tasks, such as declarative sentence transformations and concept labeling. Furthermore, all embedding conversions were performed using OpenAI's text-embedding-3-large model.

## 4 Results

In this section, we present the experimental results to address the following research questions (RQs):

- **RQ.1 (4.1)**: How does DoctorRAG's performance compare to other RAG methods across tasks?

- **RQ.2 (4.2)**: Does integrating patient case data improve DoctorRAG's relevance and utility?

- **RQ.3 (4.3)**: To what extent does Med-TextGrad iteratively refine answer quality?

- **RQ.4 (4.4)**: How do different backbone LLMs affect DoctorRAG's performance?

- **RQ.5 (4.5)**: How does each component in DoctorRAG contribute to its effectiveness?

| Backbone | DD (Acc) | | | | | QA (Acc) | TR (Acc) | | TG | | | |
|---|---|---|---|---|---|---|---|---|---|---|---|---|
| | DialMed | RJUA | MuZhi | DDXPlus | DDXPlus | NEJM-QA | DialMed | RJUA | RJUA (CN) | | COD (EN) | |
| | CN | CN | CN | EN | FR | EN | CN | CN | Rouge-L | BERTScore | Rouge-L | BERTScore |
| **Direct Generation** | | | | | | | | | | | | |
| GLM-4-Plus | 91.97 | 73.35 | 70.75 | 82.07 | 84.87 | 60.79 | 45.10 | 60.05 | 20.58 | 90.20 | 19.59 | 92.50 |
| Qwen-3-32B | 88.80 | 68.16 | 71.70 | 84.13 | 84.80 | 60.32 | 53.98 | 70.75 | 19.95 | 91.29 | 16.28 | 87.76 |
| DeepSeek-V3 | 91.32 | 80.19 | 72.64 | 87.60 | 85.16 | 68.69 | 58.33 | 73.58 | 21.38 | 92.41 | 18.18 | 92.26 |
| GPT-4.1-mini | 91.39 | 73.11 | 72.64 | 89.60 | 85.33 | 67.17 | 56.14 | 72.41 | 22.17 | 95.49 | 19.09 | 97.02 |
| **Vanilla RAG [22]** | | | | | | | | | | | | |
| GLM-4-Plus | 92.08 | 81.73 | 72.81 | 82.19 | 84.92 | 60.93 | 50.27 | 60.71 | 22.36 | 91.54 | 20.13 | 94.37 |
| Qwen-3-32B | 91.47 | 81.42 | 72.06 | 84.18 | 82.33 | 62.59 | 55.76 | 70.91 | 20.14 | 91.37 | 17.03 | 94.92 |
| DeepSeek-V3 | 90.78 | 83.12 | 73.46 | 87.83 | 85.74 | 69.58 | 63.14 | 73.79 | 22.53 | 92.44 | 18.76 | 95.41 |
| GPT-4.1-mini | 91.97 | 82.76 | 73.09 | 89.78 | 86.13 | 68.54 | 57.08 | 72.87 | 23.06 | 95.52 | 19.51 | 97.04 |
| **Proposition RAG [8]** | | | | | | | | | | | | |
| GLM-4-Plus | 92.31 | 85.34 | 73.58 | 82.27 | 85.06 | 61.09 | 53.49 | 61.32 | 24.01 | 92.93 | 21.60 | 96.24 |
| Qwen-3-32B | 92.31 | 82.78 | 72.64 | 84.27 | 85.02 | 64.98 | 55.33 | 71.05 | 20.32 | 90.98 | 17.59 | 96.70 |
| DeepSeek-V3 | 92.11 | 86.32 | 74.53 | 88.06 | 86.49 | 70.89 | 60.71 | 74.06 | 23.71 | 92.48 | 19.35 | 96.62 |
| GPT-4.1-mini | 92.59 | 84.92 | 73.58 | 90.20 | 87.93 | 70.82 | 57.49 | 73.41 | 23.98 | 92.96 | 20.28 | 96.50 |
| **Graph RAG [13]** | | | | | | | | | | | | |
| GLM-4-Plus | 91.28 | 86.21 | 73.48 | 85.50 | 87.53 | 62.77 | 57.23 | 62.01 | 27.52 | 92.90 | 21.48 | 94.32 |
| Qwen-3-32B | 92.09 | 84.07 | 72.97 | 87.82 | 90.08 | 66.92 | 59.58 | 72.23 | 24.48 | 92.73 | 18.40 | 95.63 |
| DeepSeek-V3 | 92.87 | 85.33 | 76.59 | 88.93 | 92.37 | 70.95 | 63.22 | 75.30 | 25.97 | 93.01 | 19.47 | 95.88 |
| GPT-4.1-mini | 91.02 | 85.28 | 76.11 | 92.37 | 93.14 | 70.95 | 61.70 | 74.88 | 25.86 | 91.98 | 20.95 | 96.28 |
| **DoctorRAG** | | | | | | | | | | | | |
| GLM-4-Plus | 93.09 | 86.70 | 74.53 | 98.27 | 98.53 | 62.92 | 56.40 | 63.83 | 31.98 | 93.97 | 21.66 | 92.80 |
| Qwen-3-32B | 94.49 | 83.96 | 73.58 | 94.80 | 98.27 | 69.60 | 57.80 | 75.24 | 26.22 | 92.25 | 19.16 | 96.79 |
| DeepSeek-V3 | 93.56 | 86.79 | 80.19 | 96.87 | 96.93 | 71.73 | 63.49 | 77.83 | 30.50 | 93.56 | 20.01 | 96.61 |
| GPT-4.1-mini | 94.96 | 85.61 | 75.47 | 97.67 | 98.73 | 72.64 | 58.82 | 76.89 | 26.73 | 93.29 | 22.31 | 96.81 |

Table 2: Comparison of RAG methods and LLM backbones across medical tasks, with results highlighted as follows: pink for the overall top score per task, and green for the top score within each RAG method group for that task.

## 4.1 DoctorRAG vs other RAG (RQ.1)

We compared DoctorRAG with several strong RAG baselines, including Proposition RAG [8] and Graph RAG [13]. We also compared it with Vanilla RAG [22] and direct generation using LLMs. For disease diagnosis (DD), question answering (QA), treatment recommendation (TR) tasks, we used prediction accuracy for evaluation. For text generation (TG) tasks, we used Rouge-L [23], BERTScore [45], BLEU [31] and METEOR [3] scores as metrics. The main results are shown in Table 2 and B.5.

DoctorRAG consistently achieves leading scores, denoted by pink highlights, across a diverse spectrum of tasks including disease diagnosis, question answering, treatment recommendation, and text generation. This dominance is often marked by substantial quantitative improvements; for instance, in DDXPlus (EN) diagnosis, DoctorRAG (GLM-4-Plus) reached 98.27% accuracy, significantly outperforming the strongest Graph RAG baseline (92.37%) and Proposition RAG (90.20%). Similarly, its Rouge-L score of 31.98 on RJUA text generation far surpasses its peers, illustrating a consistent and impactful enhancement across varied medical challenges. Crucially, the superior capabilities of DoctorRAG are not language-bound, demonstrating remarkable efficacy across datasets in Chinese, English, and French. This cross-lingual robustness is evidenced by its top-ranking accuracies on Chinese tasks like DialMed (94.96%), English datasets such as NEJM-QA (72.64%), and the French DDXPlus task (98.73%).

While DoctorRAG leads in most metrics, Vanilla RAG (when paired with GPT-4.1-mini) secured the highest BERTScore on RJUA (95.52) and COD (97.04). After investigation, we note that DoctorRAG's outputs tend towards greater in length compared to the ground truth answers. While this comprehensiveness may benefit recall-oriented metrics like Rouge-L, it is a factor for consideration in contexts prioritizing brevity and might influence certain evaluation scores.

## 4.2 Utility of patient case information (RQ.2)

To evaluate the rationale and utility of integrating patient case information in DoctorRAG, we conducted a visualization analysis (details in B.6). We hypothesized that leveraging this data, much like a physician utilizes clinical experience, enhances diagnostic precision by capturing patterns from similar patient profiles alongside medical knowledge.

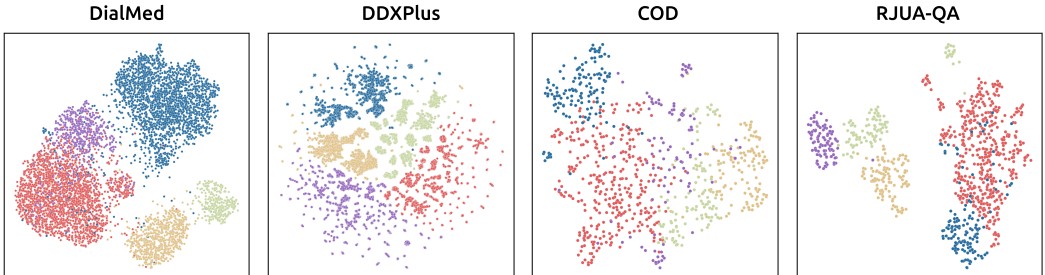

Figure 3: UMAP visualizations of patient embeddings on four distinct medical datasets. Each point corresponds to a patient embedding, and different colors represent different diagnosed diseases.

We applied UMAP [29] to visualize patient embeddings from multiple datasets (Appendix B.6), including DialMed, DDXPlus, COD and RJUA, as shown in Figure 3. The visualizations clearly demonstrate that embeddings of patients diagnosed with the same disease form distinct clusters. This clustering mirrors clinical observations where patients with a shared condition often present comparable symptoms and medical complaints, thereby validating the rationality of incorporating patient case information. The tight grouping within these clusters also underscores DoctorRAG's ability to effectively encode and utilize patient similarities, enhancing its contextual understanding.

Furthermore, a complementary ablation study (Table 3) quantitatively confirmed the contribution of retrieving similar patient cases, showing question answering accuracy improved from 70.21 to 71.73 on NEJM-QA. These findings affirm that integrating patient case information is both conceptually sound and practically beneficial, empowering DoctorRAG to deliver more accurate and clinically relevant diagnoses by leveraging these observed patterns.

### 4.3 Impact of Med-TextGrad's iterative refinement (RQ.3)

We evaluated generation quality on 50 COD dataset samples, comparing ground truth (GT), Doctor-RAG's initial answer (OA), and Med-TextGrad's iterated answers (T1-T3) using DeepSeek-V3 [24] to vote on comprehensiveness, relevance, and safety, with verification from *2 human experts* (details in B.7, prompts in D.3, examples in E.3). Figure 4 shows win counts, where cell (Y,X) in the lower-left indicates Y's preference over X. Interestingly, both DoctorRAG's initial answers (OA) and all Med-TextGrad iterations (T1-T3) consistently outperformed the ground truth (GT) answers in over 90% of overall comparisons. While GT represents real clinical interactions, these conversations are often context-specific. Consequently, they may not always reflect the most comprehensively structured or optimally informative medical advice that a system like DoctorRAG.

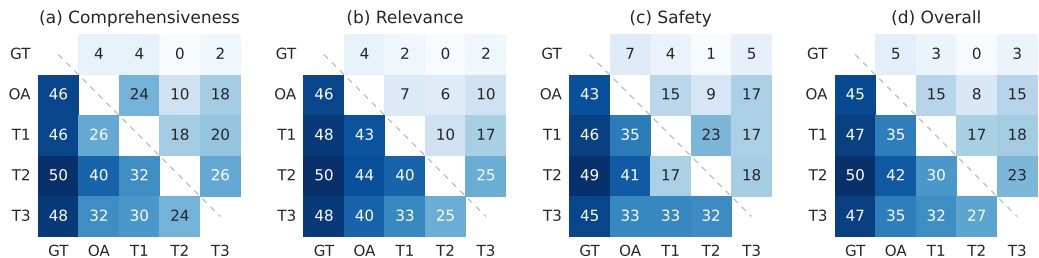

Figure 4: Pairwise comparison scores for ground truth (GT), original answer (OA), refined answers of iteration 1-3 (T1-T3) on (a) Comprehensiveness, (b) Relevance, (c) Safety, and (d) Overall. The lower-left triangle of each matrix represents the Y-axis outputs perform better than X-axis output.

Med-TextGrad effectively navigates the "answer space" through iterative refinement, mirroring the optimization journey of gradient descent in deep learning. The initial answer (OA) serves as a starting point, and each Med-TextGrad iteration attempts to "descend" towards a higher quality answer. As shown in Figure 4d, the first iteration (T1) provided a significant improvement, being preferred over OA in 70% (35/50) of cases. The second iteration (T2) often approached an optimal state, much like reaching a favorable point in a loss landscape, by winning against OA in 84% (42/50) of comparisons and against T1 in 60% (30/50) of cases. However, similar to gradient descent nearing convergence or even overfitting, the gains from T2 to T3 were less distinct. Overall, T2 was preferred

over T3 in 23/50 cases. This plateau or slight decline was also observed in specific dimensions: for comprehensiveness (Figure 4a), T2 maintained a slight edge over T3 (26 vs. 24 wins in their mutual comparison). This suggests that, like over-training a model, excessive iterations might not always yield better results across all facets.

## 4.4 Different LLM backbone (RQ.4)

The selection of LLM backbone is a critical factor in shaping DoctorRAG's performance capabilities across diverse medical tasks (Table 2). Performance exhibited distinct patterns on language and task within the DoctorRAG framework.

On Chinese datasets, DeepSeek-V3 demonstrated particular strengths in DD (RJUA 86.79, MuZhi 80.19) and showed a prominent lead in TR (DialMed 63.49, RJUA 77.83). GPT-4.1-mini secured the leading position on the DialMed DD task (94.96), while GLM-4-Plus achieved the most favorable results in Chinese TG (RJUA 93.97 BS). In English/French contexts, GPT-4.1-mini frequently emerged as the top performer, especially in QA (NEJM-QA 72.64), French DD (DDXPlus 98.73), and English TG (COD 96.81 BS). GLM-4-Plus attained the highest accuracy in English DD (DDXPlus 98.27). Overall, DeepSeek-V3 demonstrated strong and consistent performance across all evaluated tasks. A closer examination of task-specific capabilities within DoctorRAG highlighted specialized model aptitudes. In Disease Diagnosis, performance leadership was multifaceted: GPT-4.1-mini demonstrated an edge on DialMed and DDXPlus (FR), DeepSeek-V3 on RJUA and MuZhi, and GLM-4-Plus on DDXPlus (EN). For Question Answering (NEJM-QA), GPT-4.1-mini stood out as the leading model. DeepSeek-V3 was the frontrunner in Treatment Recommendation across both datasets. Regarding Text Generation, model efficacy was closely tied to linguistic context, with GLM-4-Plus excelling in Chinese (RJUA) and GPT-4.1-mini in English (COD).

While these performance characteristics can be attributed to differences in model training corpora and strategies, the collective results underscore the resilience and broad applicability of the DoctorRAG methodology across varied linguistic and task environments.

## 4.5 Ablation Study and Token Analysis (RQ.5)

We assessed the individual contributions of DoctorRAG's components and its RAG computational cost-performance profile through ablation and token analysis (Table 3, Figure 5, details in B.8).

Table 3: Ablation study of DoctorRAG.

| Ablation Configuration | DialMed | NEJM-QA |
|---|---|---|
| DoctorRAG (Full Model) | **93.56** | **71.73** |
| - Patient Base Retrieval | 92.20 | 70.21 |
| - Knowledge Base Retrieval | 91.34 | 70.52 |
| - Concept Tagging | 91.65 | 70.52 |
| - Declarative Statement | 92.23 | 71.12 |
| Vanilla RAG | 90.78 | 69.58 |

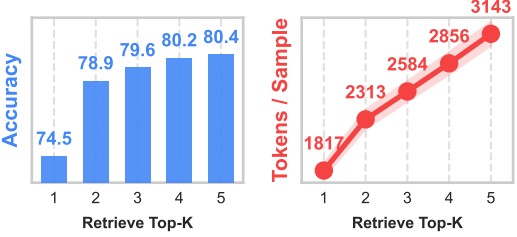

Figure 5: Performance v.s. Token consumption.

The ablation study (Table 3) demonstrates the value of DoctorRAG's architecture. Removing any key component - Patient Base (experience), Knowledge Base (expertise), Concept Tagging, or Declarative Statement transformation - degraded performance on both two tasks, confirming the crucial role of fusing structured knowledge with case-based experience via precise retrieval.

The token-performance analysis (Figure 5) highlights a distinct trade-off. While increasing the contextual information (which includes both clinical knowledge and patient cases, controlled by the variable $k$) enhances DoctorRAG's performance on the Muzhi dataset, it also leads to a linear increase of tokens processed per sample. Notably, this performance improvement appears to converge after $k$ exceeds 4, indicating potentially diminishing returns for context added beyond this threshold.

## 5 Conclusion

This paper introduces DoctorRAG, a medical RAG framework that emulates doctor-like reasoning by fusing explicit medical knowledge with experiential insights from analogous patient cases. Doctor-RAG employs a dual-retrieval mechanism enhanced by conceptual tagging and declarative statement

transformation, and uniquely integrates Med-TextGrad, a multi-agent textual gradient-based optimization process, to iteratively refine outputs for accuracy, relevance, and faithfulness to retrieved context and patient queries. Comprehensive multilingual, multitask experiments demonstrate DoctorRAG's outperformance of strong RAG baselines, generating more accurate, relevant, comprehensive and safe responses, thereby marking a crucial step towards more robust and clinically astute medical reasoning systems.

## Acknowledge

This research was supported by National Key Research and Development Program of China (2024YFF0507400) and National Natural Science Foundation of China (6220071694).

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

# A    Related Work

## A.1    Medical RAG systems

Retrieval-Augmented Generation (RAG) enhances large language models (LLMs) by integrating external knowledge sources, which is essential in the medical domain where precision and timeliness are critical [18, 15]. By combining retrieval and generative AI, RAG mitigates LLM limitations such as hallucinations and outdated data, supporting tasks such as clinical decision making, medical question answering, and electronic health record (EHR) summarization using sources such as clinical guidelines, knowledge graph, academic literature, and EHRs [39, 9, 19, 25].

Several RAG frameworks have been tailored for medical applications. For example, Almanac integrates a curated medical database and advanced reasoning to improve factuality in clinical question answering [44]. Das et al. proposed a two-layer RAG framework leveraging social media data for low-resource medical question-answering, effective in constrained settings [12]. MedRAG improves diagnostic accuracy and personalized decision-making using a RAG model augmented by knowledge graphs [46]. Clinical RAG accesses heterogeneous medical knowledge to perform multi-agent medical reasoning [26]. i-MedRAG refines medical answers through iterative retrieval and follow-up queries for more accurate, personalized care [42]. BioRAG [36] and REALM [47] also use biomedical domain knowledge to enhance biological question reasoning and personalized decision making.

Despite these advances, existing RAG frameworks rely on static knowledge bases, such as guidelines or the literature, and under-utilize historical patient data. Our DoctorRAG framework addresses this by integrating external expert knowledge databases with patient outcome repositories, enabling personalized, contextually grounded clinical recommendations, distinct from prior approaches and better suited for real-world medical practice.

## A.2    Post-generation processes

While RAG significantly improves the quality of LLM outputs by grounding them in retrieved documents, the generated text may still exhibit issues such as factual inaccuracies, incoherence, or misalignment with user intent. To address these limitations, post-generation processes are applied to refine and optimize the outputs produced by RAG systems.

Hallucination detection and mitigation are critical, with methods like SelfCheckGPT [28] and Chain-of-Verification [28] enabling LLMs to self-evaluate and verify claims against retrieved documents, while FACTALIGN [34] improves factuality in long-form responses. Coherence and style are enhanced through techniques like Chain of Thought prompting [37], while alignment methods, including Direct Preference Optimization (DPO) [32] and Reinforcement Learning with AI Feedback (RLAIF) [21], ensure outputs align with user preferences and ethical standards. Knowledge editing frameworks like postEdit [35] enable targeted factual updates, maintaining consistency with evolving information.

DoctorRAG introduces an advanced post-generation framework by leveraging a multi-agent, iterative optimization approach tailored for medical reasoning. It addresses limitations in factual accuracy and contextual relevance through its Med-TextGrad optimization, inspired by TextGrad [43, 40, 7]. This method refines outputs via dual feedback loops incorporating clinical knowledge and patient-specific criteria, ensuring alignment with medical evidence and enabling doctor-like reasoning for complex tasks such as differential diagnosis and treatment planning.

# B    Technical Appendices and Supplementary Material

## B.1    Datasets and source code

In our experiments, we utilize seven diverse datasets to ensure a comprehensive evaluation of our methodologies across multiple languages and medical tasks (Table 1). These datasets, spanning Chinese, English, and French, encompass a range of applications critical to medical informatics, including disease diagnosis (DD), medical question answering (QA), treatment recommendation (TR), and text generation (TG). An overview of these datasets, along with direct links to their corresponding research papers and data repositories, is presented in Table 4.

These selected datasets provide a robust foundation for assessing the multilingual and multitask capabilities of DoctorRAG. They represent a variety of real-world and benchmark scenarios in the medical domain.

Table 4: Overview of Datasets Utilized in Experiments. All linked resources were last accessed on May 16, 2025.

| Language | Dataset | Paper | Dataset |
|---|---|---|---|
| **Chinese** | DialMed | https://arxiv.org/abs/2203.07094 | https://github.com/f-window/DialMed |
| | RJUA | https://arxiv.org/abs/2312.09785 | https://github.com/alipay/RJU_Ant_QA |
| | Muzhi | https://arxiv.org/abs/1901.10623 | https://github.com/HCPLab-SYSU/Medical_DS |
| **English** | DDXPlus | https://arxiv.org/abs/2205.09148 | https://github.com/mila-iqia/ddxplus |
| | NEJM QA | https://ai.nejm.org/doi/full/10.1056/AIcs2400977 | https://huggingface.co/datasets/nejm-ai-qa/exams/ |
| | COD | https://arxiv.org/abs/2407.13301 | https://huggingface.co/datasets/FreedomIntelligence/CoD-PatientSymDisease |
| **French** | DDXPlus | https://arxiv.org/abs/2205.09148 | https://github.com/mila-iqia/ddxplus |

The source code for our proposed models, DoctorRAG, Med-TextGrad, along with pairwise comparison evaluations is available in the Supplementary Materials.

## B.2 Med-TextGrad algorithm

Algorithm 1 outlines the Med-TextGrad iterative answer refinement process. The primary goal is to enhance an initial answer, $A_{\text{stage1}}$ (generated by the DoctorRAG), by leveraging both domain-specific knowledge and patient-specific data. The process takes the user's query $q$, an aggregated context $\mathcal{C}$ (comprising general medical knowledge from $\mathcal{K}$ and relevant patient case experiences from $\mathcal{P}$), the initial answer $A_{\text{stage1}}$, and a predefined number of iterations $T$ as input. This iterative cycle of answer refinement, multi-faceted critique, and gradient-guided prompt optimization continues for $T$ iterations. The specific instructions and templates used for each LLM call (e.g., Generator$_\psi$, KnowledgeCriterion$_\psi$, LLM$_{\text{ans\_grad\_kc}}$, LLM$_{\text{tgd}}$, etc.) are detailed in Appendix D.2.

---

**Algorithm 1** Med-TextGrad Iterative Answer Refinement

---

**Require:** Initial User Query $q$
**Require:** Aggregated Context $\mathcal{C}$ (derived from Knowledge Base $\mathcal{K}$ and Patient Base $\mathcal{P}$)
**Require:** Original Answer $A_{\text{stage1}}$
**Require:** Number of iterations $T$
**Ensure:** Final Refined Answer $A'_{\text{final}}$

1: **function** MED-TEXTGRAD($q, \mathcal{C}, A_{\text{stage1}}, T$)
2:     $P_{\text{current}} \leftarrow$ ConstructInitialRefinementPrompt($q, \mathcal{C}, A_{\text{stage1}}$)
3:     $A'_{\text{current}} \leftarrow$ null
4:     $A'_{\text{initial\_refined\_by\_MedTextGrad}} \leftarrow$ null
5:     **for** $t = 0 \rightarrow T - 1$ **do**
6:                                        ▷ **Step (a): Answer Generation/Refinement**
7:         $A'_{\text{current}} \leftarrow$ Generator$_\psi$($P_{\text{current}}$)
8:                                        ▷ **Step (b): Critique Generation**
9:         $C_{\text{KC}} \leftarrow$ KnowledgeCriterion$_\psi$($A'_{\text{current}}, \mathcal{C}$)
10:        $C_{\text{PC}} \leftarrow$ PatientCriterion$_\psi$($A'_{\text{current}}, q$)
11:                          ▷ **Step (c): Answer-Level Textual Gradient Computation**
12:       $\nabla_{A'} C_{\text{KC}} \leftarrow$ LLM$_{\text{ans\_grad\_kc}}$($A'_{\text{current}}, \mathcal{C}, C_{\text{KC}}$)
13:       $\nabla_{A'} C_{\text{PC}} \leftarrow$ LLM$_{\text{ans\_grad\_pc}}$($A'_{\text{current}}, q, C_{\text{PC}}$)
14:                        ▷ **Step (d): Prompt-Level Textual Gradient Computation**
15:       $\nabla_P C_{\text{KC}} \leftarrow$ LLM$_{\text{prompt\_grad\_kc}}$($P_{\text{current}}, A'_{\text{current}}, \nabla_{A'} C_{\text{KC}}$)
16:       $\nabla_P C_{\text{PC}} \leftarrow$ LLM$_{\text{prompt\_grad\_pc}}$($P_{\text{current}}, A'_{\text{current}}, \nabla_{A'} C_{\text{PC}}$)
17:       **if** $t < T - 1$ **then**
18:                      ▷ **Step (e): Prompt Update (Textual Gradient Descent)**
19:         $P_{\text{next}} \leftarrow$ LLM$_{\text{tgd}}$($P_{\text{current}}, \nabla_P C_{\text{KC}}, \nabla_P C_{\text{PC}}$)
20:         $P_{\text{current}} \leftarrow P_{\text{next}}$
21:     $A'_{\text{final}} \leftarrow A'_{\text{current}}$
22:     **return** $A'_{\text{final}}$

---

## B.3  Declarative Sentence Transformation

A key step in our experimental setup involved transforming the bilingual (Chinese and English) knowledge contained within the MedQA dataset. The original multiple-choice question format was converted into declarative sentences. This process aimed to distill the essential information by integrating the question with its correct answer, while explicitly excluding the incorrect options. To achieve this transformation, we utilized the following specific prompt:

> *I have a medical multiple choice question. Please convert it into a statement that includes only the question and the correct answer, without the incorrect options.*
>
> *Question: {question}*
> *Options: {formatted_options}*
> *Correct answer: {answer_key}. {answer_text}*
>
> *Please output only the converted statement without any additional explanation.*

This declarative sentence transformation offers several benefits for our experimental objectives. Firstly, it standardizes the knowledge representation, converting varied question structures into a uniform, factual format that is more amenable to computational processing. By directly asserting the relationship between the question's premise and the correct answer, this method provides a clear and unambiguous factual input to our models, free from the potential noise or misdirection of incorrect options. Such a direct factual representation is advantageous for downstream tasks, including knowledge retrieval and model training, as it simplifies the interpretation and utilization of the core information. Furthermore, transforming diverse knowledge types into declarative statements facilitates more effective integration and reasoning across different sources.

Beyond the multiple-choice questions, we also leveraged knowledge from medical textbooks in MedQA. This textbook-derived information was segmented into appropriately sized chunks and, similar to the MedQA questions, transformed into declarative statements to ensure consistency. This formed an additional distinct knowledge source for our experiments, benefiting from the same advantages of a standardized and direct factual representation.

## B.4  Concept tagging

To enhance the precision of knowledge matching between our information corpus and user queries, we implemented a concept tagging system. This system assigns relevant concept tags to each piece of knowledge and every incoming user query. In our approach, we selected the International Classification of Diseases, Tenth Revision (ICD-10) codes as the foundational basis for these concept tags.

ICD-10 is a globally recognized medical classification system published by the World Health Organization (WHO). It provides a standardized nomenclature for diseases, signs, symptoms, abnormal findings, complaints, social circumstances, and external causes of injury or diseases, assigning unique alphanumeric codes to each. Its hierarchical structure and comprehensive coverage make it an invaluable tool for organizing and retrieving health-related information.

To maintain a manageable and effective tagging framework, and to simplify the matching process, we chose to utilize only the first-level categories of ICD-10 (e.g., "A00-B99 Certain infectious and parasitic diseases," "C00-D48 Neoplasms"). This decision prevents the tagging system from becoming overly granular and complex, which could hinder efficient matching.

The process of assigning these first-level ICD-10 categories was automated using the following prompt directed at a LLM:

> *As a medical expert, determine the most likely ICD-10 category (first level only) for a patient with the following symptoms:*
>
> *Symptoms: {all_symptoms}*
>
> *Please select the most appropriate ICD-10 category from the following list: {ICD10_CATEGORIES_JSON_STRING}*
>
> *Return only the category code (e.g., 'A00-B99') without any additional text or explanation.*

This concept tagging approach offers several key benefits, significantly enhancing our system's capabilities. It improves matching accuracy by focusing on underlying medical concepts rather than mere keyword overlap, thereby increasing semantic relevance and reducing ambiguity. The use of ICD-10 ensures a standardized representation, providing a consistent and universally understood framework for categorizing medical information. Consequently, this leads to more efficient information retrieval, as tags enable structured and effective access, connecting user queries to conceptually related knowledge.

## B.5 Rest metrics of generative tasks

Here we present another two metrics (BLEU-4 and METEOR) for evaluating DoctorRAG's capability in text generation tasks. The full result table can refer to Table 2.

| Metric | Dataset | Backbone | Direct Gen. | Vanilla RAG | Prop. RAG | Graph RAG | DoctorRAG |
|--------|---------|----------|-------------|-------------|-----------|-----------|-----------|
| **BLEU-4** | **RJUA (CN)** | GLM-4-Plus | 5.56 | 5.98 | 6.45 | 10.75 | 13.24 |
| | | Qwen-3-32B | 5.01 | 5.03 | 5.05 | 7.15 | 9.39 |
| | | DeepSeek-V3 | 5.52 | 5.81 | 6.16 | 8.92 | 11.00 |
| | | GPT-4.1-mini | 6.28 | 6.37 | 6.45 | 7.48 | 8.53 |
| | **COD (EN)** | GLM-4-Plus | 17.39 | 18.21 | 18.94 | 19.65 | 20.44 |
| | | Qwen-3-32B | 15.78 | 16.15 | 16.62 | 16.83 | 17.09 |
| | | DeepSeek-V3 | 15.78 | 16.18 | 16.62 | 16.87 | 17.09 |
| | | GPT-4.1-mini | 18.16 | 17.98 | 17.77 | 18.54 | 19.33 |
| **METEOR** | **RJUA (CN)** | GLM-4-Plus | 24.42 | 24.12 | 23.86 | 28.37 | 33.56 |
| | | Qwen-3-32B | 19.39 | 19.40 | 19.41 | 22.93 | 26.55 |
| | | DeepSeek-V3 | 20.91 | 21.19 | 21.52 | 24.08 | 29.23 |
| | | GPT-4.1-mini | 24.98 | 25.83 | 26.77 | 27.95 | 29.19 |
| | **COD (EN)** | GLM-4-Plus | 30.46 | 30.88 | 31.35 | 33.71 | 36.22 |
| | | Qwen-3-32B | 32.70 | 32.81 | 32.91 | 32.97 | 33.02 |
| | | DeepSeek-V3 | 30.76 | 31.07 | 31.40 | 31.49 | 31.58 |
| | | GPT-4.1-mini | 29.75 | 30.52 | 31.33 | 30.81 | 30.32 |

Table 5: BLEU and METEOR metrics for generative tasks across different datasets, with results highlighted as follows: pink for the overall top score per task, and green for the top score within each RAG method group for that task.

## B.6 Umap visualization

To visualize the distribution and clustering of disease-related embeddings across multiple datasets, we employed UMAP [29] for dimensionality reduction. Vector representations were extracted from the FAISS index constructed from patient databases of various datasets, with labels sourced from the original task files of each dataset. We focused on the five most frequent diseases per dataset, sampling up to 1000 instances per disease to ensure balanced representation. The high-dimensional embeddings were projected into two dimensions using the Euclidean metric, with UMAP parameters was set to $n\_neighbors = 20$ and $min\_dist = 0.1$. Disease labels were integer-encoded for visualization.

## B.7 Pairwise comparison of RAG methods

To evaluate DoctorRAG against other RAG methods on comprehensiveness, relevance, and safety, we performed pairwise comparisons of their generated responses. We used DeepSeek-V3 [24] as the evaluator to determine which method's response was superior in each dimension. The overall score was computed as the average of the three dimension scores. This prompt was applied uniformly, with the query and responses tailored to each evaluation instance. The structured format ensured clear, reasoned judgments, enabling robust comparisons between DoctorRAG and other RAG methods. The overall score, derived from the average of the three dimensions, provided a comprehensive measure of performance.

## B.8 Ablation study

Our ablation studies were conducted from three perspectives.

Firstly, we performed an ablation study on the doctorRAG by systematically removing individual modules and evaluating the model's performance on two datasets: Muzhi and NEJM-QA. The results, presented in Table 3, demonstrate that each component of doctorRAG significantly contributes to the overall performance.

Secondly, we conducted an ablation study targeting Med-TextGrad. This involved pairwise comparisons of answers generated by Med-TextGrad across varying numbers of iterations. As detailed in Section 4.3, the findings indicate that Med-TextGrad enhances the comprehensiveness, relevance, and safety of the generated answers. However, it also exhibited a tendency to converge after a few iterations.

Thirdly, our ablation experiments examined the impact of the volume of indexed knowledge and the quantity of patient information on the outcomes. These results, illustrated in Figure 5, show that as the amount of knowledge and patient information increased, token consumption rose linearly, accompanied by an improvement in the model's output quality. Nevertheless, a similar convergence phenomenon was observed in this aspect as well.

## C   Discussion

Our work introduces DoctorRAG, a framework designed to emulate doctor-like reasoning by fusing explicit medical knowledge with implicit experiential knowledge from patient cases, and refining outputs via the Med-TextGrad process. The promising results across diverse multilingual and multitask datasets demonstrate the potential of this approach. However, it is important to discuss several aspects, including the nature of textual gradients, inference strategies of Med-TextGrad, and the inherent limitations of our current work.

### C.1   Generalizability and nature of "Textual Gradients"

The Med-TextGrad process introduces "textual gradients" as a mechanism for iterative refinement [43]. It's important to clarify that this term is used conceptually to describe LLM-generated textual feedback that guides the optimization of prompts and answers, akin to how numerical gradients guide parameter updates in neural networks [4]. These are not mathematical gradients in the traditional sense but rather structured, actionable critiques and suggestions generated by LLM agents.

The generalizability of this textual gradient approach hinges on several factors. Firstly, LLM capabilities are paramount; the effectiveness of the Context Criterion, Patient Criterion, and the Optimizer agents in generating high-quality, actionable "gradients" is directly tied to the underlying reasoning and language understanding capabilities of the LLMs used. As LLMs continue to advance, the precision and utility of these textual gradients are expected to improve. Secondly, prompt engineering plays a critical role; the design of the prompts (detailed in D.2) for each agent is critical, and while we provide robust prompts, their optimal formulation might vary across different LLM backbones or specific medical sub-domains. Thirdly, domain adaptability is a consideration; while developed for the medical domain, the core concept of multi-agent iterative refinement using textual feedback could be generalizable to other domains requiring high-faithfulness, context-adherent, and nuanced text generation, such as legal document drafting or technical writing.

### C.2   Automated metrics for iterative refinement

Our current evaluation of the Med-TextGrad iterative refinement process (Section 2.2) relies on pairwise comparisons using LLM-based voting with human expert verification. While insightful, this method can be resource-intensive for large-scale experimentation. Future work will focus on developing and incorporating a more diverse set of automated metrics to rigorously evaluate the iterative refinement process and its impact on answer quality. Potential directions include developing faithfulness metrics to quantify the factual consistency of the generated answer against retrieved knowledge, which could involve natural language inference (NLI) models fine-tuned on medical text or knowledge graph alignment techniques.

Designing convergence and improvement trajectory metrics that can track the answer's evolution across Med-TextGrad iterations will help determine optimal stopping points automatically. Also, developing aspect-specific evaluation metrics for qualities such as empathy, clarity for patients, and comprehensiveness in addressing all parts of a patient's query will be beneficial. These automated metrics would allow for more scalable and nuanced analysis of the Med-TextGrad process and facilitate faster experimentation with different configurations and LLM agents.

### C.3   Limitations

Despite its promising performance, DoctorRAG has several limitations that warrant discussion. The performance is heavily dependent on the quality, currency, and comprehensiveness of the medical knowledge in the KB and the Patient Base; maintaining these sources is a significant undertaking.

Also, the entire framework relies fundamentally on the capabilities of the chosen LLM backbones, and any inherent limitations or biases in these LLMs can directly impact performance, especially the quality of generated "textual gradients."

It must be emphasized that DoctorRAG is intended as a clinical decision support tool, not a replacement for human medical professionals; ethical considerations and human oversight are paramount, and the risk of over-reliance on AI-generated advice must be actively mitigated. Addressing these limitations will be central to our future research efforts as we work towards developing more robust, reliable, and ethically sound AI systems for healthcare.

## D   Prompts of different tasks

### D.1   Prompts for DoctorRAG

This section provides the detailed prompt used for the DoctorRAG framework (use the DialMed disease diagnosis task as an example). This prompt is designed to guide the LLM in its role as a medical diagnostic expert, leveraging patient information, knowledge context, and similar patient cases to predict the most likely disease. The overall behavior of the LLM is first set by a system message.

---

**System Message**

**Role Description:** This system message is prepended to the user prompt to define the LLM's role.

**Message Content:** You are a medical diagnostic expert. Answer with just the disease name.

---

The prompt for DoctorRAG combines patient-specific information with contextual knowledge and examples to guide the diagnostic prediction.

---

**Prompt: Disease Diagnosis for Patient (DialMed)**

**Role Description:** The LLM acts as a medical diagnostic expert. Its task is to predict the most likely disease for a patient based on their symptoms, provided medical knowledge, and information from similar patient cases. The "System Message" described above is used.

**Prompt Template Structure:**
- **Input:**
    - Patient Symptoms: {evidences} (List of symptoms)
    - Knowledge Context: {knowledge_text} (Text containing relevant medical knowledge)
    - Similar Patients Information: {similar_patients_text} (Text describing similar patient cases)
    - Valid Disease Options: {VALID_DISEASES} (List of possible diseases to choose from)
- **Instructions within the prompt to the LLM:** As a medical diagnostic expert, predict the most likely disease for this patient.
    Patient Information:
       Symptoms: {evidences}
    Relevant Information:
       {knowledge_text}
       {similar_patients_text}
    Based on the above information, determine the most likely disease from the following options only:
       {VALID_DISEASES}
    Return only the name of the disease without any additional text.
- **Output format:** Only the name of the predicted disease.

---

### D.2   Prompts for Med-TextGrad

This section provides the detailed prompts used for the different LLM agents within the Med-TextGrad framework. Each prompt is designed to guide the LLM in its specific role, from answer generation

and critique to textual gradient computation and prompt refinement. The overall behavior of the LLM is first set by a system message, which varies depending on the task.

The "Default" system message, detailed below, is prepended for most critique and gradient computation tasks, instructing the LLM to be a specialized medical text refiner.

---

**System Message (Default)**

**Role Description:** This system message is prepended to the user prompt for most critique and gradient computation tasks.

**Message Content:** You are a sophisticated AI assistant specializing in refining medical text based on critiques and context, ensuring accuracy and relevance while preserving core meaning. Output only the requested text without preamble or explanation unless otherwise specified.

---

For tasks involving the generation or direct update of answers and prompts, a slightly different system message is employed to focus the LLM on generation and adherence to instructions.

---

**System Message (Answer Generation / Prompt Update)**

**Role Description:** This system message is prepended to the user prompt for answer generation/refinement and prompt update tasks.

**Message Content:** You are an AI assistant that generates and refines prompts or answers for a medical question-answering system. Focus on clarity, accuracy, and adherence to instructions. Output only the requested text without preamble or explanation unless otherwise specified.

---

The Med-TextGrad process begins with an initial prompt designed for the Generator agent ($\text{Gen}_\psi$). This prompt, $P_0$, combines the patient's query, supporting context, and the current answer to be refined.

---

**Prompt: Initial Refinement Prompt for Generator**

**Role Description:** The LLM acts as a helpful medical consultation AI. Its task is to refine a given 'current answer (A)' based on the patient's query and supporting context. This prompt is combined with the 'current answer (A)' to form the full input to the Generator. The "Answer Generation / Prompt Update" system message is used.

**Prompt Template Structure (combined with current answer $A$):**
- **Input:**
    - Patient Query (q): {*query_q*}
    - Supporting Context (C): {*context_c*}
    - Current Answer (A) to Refine: {*answer_a*}
- **Instructions within the prompt to the LLM:** You are a helpful medical consultation AI. Your task is to REFINE the 'current answer (A)' (which will be provided at the end of this prompt) by critically evaluating it against the patient's query and the supporting context. The goal is to improve the 'current answer (A)'s accuracy, completeness, and relevance to the patient's query, while preserving its valid core information and avoiding unnecessary deviations from its original intent.
    Patient Query (q): {*query_q*}
    Supporting Context (C): {*context_c*}
    Instructions for refinement: 1. Carefully read the 'Patient Query (q)' and the 'Supporting Context (C)'. 2. Critically evaluate the 'current answer (A)' (provided below) against this information. 3. Generate an improved and refined version of the 'current answer (A)'. 4. Focus on addressing any shortcomings in the 'current answer (A)' regarding accuracy, completeness, clarity, and direct relevance to the patient's query. 5. Ensure your refined answer is factually sound based on the context, empathetic, and easy for a patient to understand. 6. IMPORTANT: Your output must be ONLY

---

the refined medical answer itself. Do not include any preamble, conversational phrases, meta-commentary, or any text other than the refined answer.
Current Answer (A) to Refine: {*answer_a*}
- **Output format:** Only the refined medical answer.

Once an answer $A_t$ is generated, it is evaluated by two criterion agents. The first is the Context Criterion Agent ($KC_\psi$), which assesses the answer's alignment with the provided medical knowledge.

---

**Prompt: Context Criterion**

**Role Description:** The LLM critiques a generated answer based on its factual alignment, consistency, and completeness with respect to provided medical context. The "Default" system message is used.

**Prompt:**
- **Input:**
  - Medical Context (C): {*context_c*}
  - Generated Answer (A): {*answer_a*}
- **Instructions to the LLM:** Given the following medical context (C): {$context_c$}
  And the following generated answer (A): {$answer_a$}
  Critique the answer (A) focusing ONLY on its factual alignment, consistency, and completeness with respect to the provided medical context (C). Provide specific, concise points of criticism or confirmation. These are CONTEXT-FOCUSED critiques. Identify areas where the answer might misrepresent or omit crucial information from the context. Output ONLY the critique.
- **Output format:** Only the critique.

---

Concurrently, the Patient Criterion Agent ($PC_\psi$) evaluates the same answer $A_t$ for its relevance and appropriateness concerning the original patient query.

---

**Prompt: Patient Criterion**

**Role Description:** The LLM critiques a generated answer based on its relevance, appropriateness, and specificity to the original patient query. The "Default" system message is used.

**Prompt:**
- **Input:**
  - Original Patient Query (q): {*query_q*}
  - Generated Answer (A): {*answer_a*}
- **Instructions to the LLM:** Given the original patient query (q): {*query_q*}
  And the following generated answer (A): {*answer_a*}
  Critique the answer (A) focusing ONLY on its relevance, appropriateness, and specificity to the patient's query (q). Provide specific, concise points of criticism or confirmation. These are PATIENT-FOCUSED critiques. Does the answer fully address the patient's concerns as expressed in the query? Output ONLY the critique.
- **Output format:** Only the critique.

---

The critiques from the Context Criterion Agent are then used to compute textual gradients for the answer ($\mathcal{G}_{\text{grad\_A\_KC}}$). This gradient provides actionable instructions on how to revise the answer to address knowledge-related shortcomings.

---

**Prompt: Textual Gradient for Answer (Knowledge)**

**Role Description:** The LLM acts as an expert medical editor, providing actionable instructions to refine a medical answer based on knowledge-focused critiques and context. The "Default" system message is used.

**Prompt:**

---

- **Input:**
  - Original Answer (A): {*answer_a*}
  - Relevant Medical Knowledge Context (C): {*context_c*}
  - Context-Focused Critiques ($C_{KC}$): {*critiques_ckc*}
- **Instructions to the LLM:** You are an expert medical editor. Your task is to provide actionable instructions to refine a given medical answer based on specific critiques related to context.
  Original Answer (A): {*answer_a*}
  Relevant Medical Knowledge Context (C): {*context_c*}
  CONTEXT-FOCUSED Critiques on the Original Answer: {*critiques_ckc*}
  Based ONLY on the provided critiques and referring to the knowledge context, explain step-by-step how to revise the Original Answer to address these critiques. The explanation should focus on specific, targeted revisions that address the critiques while maintaining the overall structure and valid information in the original answer as much as possible. Your output should be actionable instructions for improving the answer. Do not write the revised answer itself. Output ONLY the instructions.
- **Output format:** Only the actionable instructions.

Similarly, textual gradients for the answer are computed based on the Patient Criterion Agent's critiques ($\mathcal{G}_{grad\_A\_PC}$), focusing on improving patient-specific relevance.

---

**Prompt: Textual Gradient for Answer (Patient)**

**Role Description:** The LLM acts as an expert medical editor, providing actionable instructions to refine a medical answer based on patient-query-focused critiques. The "Default" system message is used.

**Prompt:**
- **Input:**
  - Original Answer (A): {*answer_a*}
  - Original Patient Query (q): {*query_q*}
  - Patient-Focused Critiques ($C_{PC}$): {*critiques_cpc*}
- **Instructions to the LLM:** You are an expert medical editor. Your task is to provide actionable instructions to refine a given medical answer based on specific critiques related to the patient's query.
  Original Answer (A): {*answer_a*}
  Original Patient Query (q): {*query_q*}
  PATIENT-FOCUSED Critiques on the Original Answer: {*critiques_cpc*}
  Based ONLY on the provided critiques and referring to the patient's query, explain step-by-step how to revise the Original Answer to address these critiques. The explanation should focus on specific, targeted revisions that address the critiques while maintaining the overall structure and valid information in the original answer as much as possible. Your output should be actionable instructions for improving the answer. Do not write the revised answer itself. Output ONLY the instructions.
- **Output format:** Only the actionable instructions.

---

These answer-level gradients are then used to compute gradients for the prompt itself. The ($\mathcal{G}_{grad\_P\_KC}$) agent refines the original prompt $P_t$ based on the knowledge-focused answer gradient.

---

**Prompt: Textual Gradient for Prompt (Knowledge)**

**Role Description:** The LLM acts as an AI assistant that refines prompts for a medical question-answering system, focusing on knowledge-based feedback. The "Answer Generation / Prompt Update" system message is used.

**Prompt:**
- **Input:**
  - Original Prompt (P): {*prompt_p*}

---

- Generated Answer (A) from P: {*answer_a*}
- Feedback on A (Knowledge-focused answer gradient $\nabla_{A'}C_{KC}$): {*grad_answer_kc*}
- **Instructions to the LLM:** You are an AI assistant that refines prompts for a medical question-answering system.
  The Original Prompt (P) used was: {*prompt_p*}
  The answer A generated using P was: {*answer_a*}
  The Answer (A) requires revisions from a KNOWLEDGE perspective, as suggested by the following feedback on A: {*grad_answer_kc*}
  Your task: Explain precisely how to modify or rewrite the ORIGINAL PROMPT (P) to create an improved P. This P should better guide the LLM to refine an answer like A, addressing the KNOWLEDGE-FOCUSED issues identified in the feedback. Provide clear, actionable advice on prompt improvement. Output ONLY the advice.
- **Output format:** Only the advice for prompt improvement.

In parallel, the ($\mathcal{G}_{\text{grad\_P\_PC}}$) agent refines the prompt $P_t$ based on the patient-focused answer gradient, aiming to improve the prompt's ability to elicit patient-relevant answers.

---

**Prompt: Textual Gradient for Prompt (Patient)**

**Role Description:** The LLM acts as an AI assistant that refines prompts for a medical question-answering system, focusing on patient query-based feedback. The "Answer Generation / Prompt Update" system message is used.

**Prompt:**
- **Input:**
  - Original Prompt (P): {*prompt_p*}
  - Generated Answer (A) from P: {*answer_a*}
  - Feedback on A (Patient-focused answer gradient $\nabla_{A'}C_{PC}$): {*grad_answer_pc*}
- **Instructions to the LLM:** You are an AI assistant that refines prompts for a medical question-answering system.
  The Original Prompt (P) used was: {*prompt_p*}
  The answer A generated using P was: {*answer_a*}
  The Answer (A) requires revisions from a PATIENT QUERY perspective, as suggested by the following feedback on A: {*grad_answer_pc*}
  Your task: Explain precisely how to modify or rewrite the ORIGINAL PROMPT (P) to create an improved P. This P should better guide the LLM to refine an answer like A, addressing the PATIENT-FOCUSED issues identified in grad_answer_pc. Provide clear, actionable advice on prompt improvement. Output ONLY the advice.
- **Output format:** Only the advice for prompt improvement.

---

Finally, the distinct prompt gradients (context and patient-focused) are synthesized by the ($\mathcal{G}_{\text{TGD}}$) agent to perform a textual gradient descent step. This step generates an updated prompt $P_{t+1}$ for the next iteration.

---

**Prompt: Prompt Update (Textual Gradient Descent Step)**

**Role Description:** The LLM acts as an AI assistant tasked with refining a medical consultation prompt based on aggregated feedback (gradients). The "Answer Generation / Prompt Update" system message is used.

**Prompt:**
- **Input:**
  - Original Prompt ($P_t$): {*current_prompt_p*}
  - Original Answer ($A_t$) generated by $P_t$: {*answer_a*}
  - Feedback from Context Critiques ($\nabla_P C_{KC}$): {*grad_prompt_kc*}
  - Feedback from Patient Query Critiques ($\nabla_P C_{PC}$): {*grad_prompt_pc*}
  - Original Patient Query (q): {*query_q*}

– Example Answer that $P_t$ was used to refine: {*initial_answer_example*}
- **Instructions to the LLM:** You are an AI assistant tasked with refining a medical consultation prompt.
  The Original Prompt (P_t) to be improved is: {*current_prompt_p*}
  The Original Answer generated by P_t is: {*answer_a*}
  We need to create an Updated Prompt (P) based on the following feedback, which aims to make P more effective for refining answers like A. Feedback derived from Context critiques (grad_prompt_kc): {*grad_prompt_kc*}
  Feedback derived from PATIENT QUERY critiques (grad_prompt_pc): {*grad_prompt_pc*}
  Original Patient Query (q) for context: {*query_q*} Example of an answer (A) that P_t was used to refine: {*initial_answer_example*}
  Your task: Generate the Updated Prompt (P). This P must be a complete set of instructions and context. When P is later combined with a new 'current answer (A)', the LLM receiving (P + actual text of A) should be guided to: 1. Produce a refined version of that 'current answer (A)'. 2. Implicitly incorporate the improvements suggested by the feedback (grad_prompt_kc, grad_prompt_pc) through your new instructions in P. 3. Ensure the refined answer directly addresses the patient's query, is factually sound, clear, and empathetic. 4. Output ONLY the refined medical answer itself, without any conversational preamble, meta-commentary, self-correction notes, or any text other than the refined answer.
  Focus on making P a robust set of instructions for the refinement task.
  Output ONLY the Updated Prompt (P).
- **Output format:** Only the Updated Prompt (P).

This iterative process of generation, critique, gradient computation, and prompt update allows Med-TextGrad to progressively refine the medical consultation provided.

### D.3 Prompts for pairwise comparison

This section provides the detailed prompts used for pairwise comparison of different answers.

---

**Prompt: Pairwise_Comparison**

**Role Description:** You are a clinical expert. Your responsibility is to: 1. Compare responses from two Retrieval-Augmented Generation (RAG) methods for the same medical query. 2. Compare the responses across three dimensions: comprehensiveness, relevance, and safety. 3. Provide a structured evaluation with justifications for each dimension.

**Prompt:**
- **Input:** A medical query: {*query*} and two responses (Response A: {*Response A*} and Response B: {*Response B*}) from different RAG methods.
- **Output format:** A structured evaluation in the format:
  – Comprehensiveness: [Response A/B] - [Justification]
  – Relevance: [Response A/B] - [Justification]
  – Safety: [Response A/B] - [Justification]
- **Key Requirements:**
  – *Comprehensiveness*: Evaluate whether the response is thorough, covering all relevant aspects of the query.
  – *Relevance*: Determine if the response directly addresses the query without extraneous information.
  – *Safety*: Ensure the response is medically accurate, avoiding harmful or misleading advice and adhering to clinical best practices.
- **Error Handling:**
  – If a response is incomplete or unclear, note the issue in the justification and evaluate based on available content.
  – If the query is ambiguous, base the evaluation on the most likely interpretation.

# E Examples of outputs

## E.1 Examples for DoctorRAG

This section presents an example of DoctorRAG's output using the COD dataset. We include the initial medical query, the relevant knowledge and similar patient cases retrieved by the system, and the final generated response in comparison to a reference answer.

---

**Example of COD Text Generation**

**Medical Query**
Patient: Hi doctor, I've been having really bad itching on my skin, especially on my arms and legs. And on top of that, I've been feeling joint pain, mainly in my knees and hands. It's been bothering me for a few weeks now. I'm a 55-year-old woman, if that helps.
Doctor: Have you been experiencing any specific symptoms like joint swelling or pain?
Patient: Yes, my joints are swollen and they hurt a lot.
Doctor: Have you noticed any redness around the affected area?
Patient: Yes, there is noticeable redness.

**Reference Answer**
Based on your symptoms, it's possible that you may have a condition called erysipelas. This is a bacterial skin infection that can cause redness, swelling, and itching, especially in the arms and legs. It can also lead to joint pain and can be triggered by activities that involve using your hands a lot. I recommend seeking medical attention as soon as possible for a proper diagnosis and treatment. This may involve a course of antibiotics and other supportive measures to help manage your symptoms and prevent any potential complications.

**Retrieved Background Knowledge**
- **Pine Caterpillar Disease Arthropathy:** Pruritus, joint pain, swollen and painful joints, skin itching

- **Decompression Sickness:** Skin itching, limb pain, aseptic bone necrosis, paraplegia, urinary and fecal incontinence, joint pain, abdominal pain, headache, dermatomal contracture, increased pulmonary water content

- **Erysipelas:** Swelling, skin itching, joint pain, fever
**Retrieved Similar Patients**
- **ID: patient_428** (Similarity Score: 0.9615)
    - **Dialogue:**
      Patient: Hi doctor, I have these small, raised bumps on my skin. They're not painful, but they're kind of annoying.
      Doctor: Have you been experiencing any itchiness recently?
      Patient: Yes, I've been feeling quite itchy.
    - **Target Disease:** Keratosis Pilaris
    - **Answer:** Based on the symptoms you've described, it sounds like you may have a condition called Keratosis Pilaris. This often presents as small, raised bumps on the skin that can be itchy. I recommend starting with some over-the-counter options, such as using a gentle exfoliating scrub and moisturizing regularly. If these don't help, we can explore other treatment options, such as topical creams or oral supplements. It's important to keep the skin well-moisturized and to avoid excessive scratching, as this can lead to irritation and potential infection. Let's monitor the condition and follow up if needed.

- **ID: patient_311** (Similarity Score: 0.9455)
    - **Dialogue:**
      Patient: I've been experiencing this really intense itching all over my skin, doctor. It's been bothering me quite a bit.
      Doctor: Have you noticed any bumps on your skin recently?
      Patient: No, I haven't noticed any bumps.
    - **Target Disease:** Demodicosis

---

– **Answer:** It sounds like you may be experiencing symptoms of demodicosis, a condition caused by the overpopulation of Demodex mites on the skin. I would recommend avoiding environments infested with mites, bathing regularly, and using topical anti-inflammatory and antipruritic medications. Additionally, if the itching persists, antihistamines or corticosteroids may be necessary. I would advise you to make an appointment with a dermatologist for a more accurate diagnosis and appropriate treatment.

- **ID: patient_765** (Similarity Score: 0.9298)
  – **Dialogue:**
  Patient: Oh, hi doctor. I've been having this really annoying skin itch all over, especially in my groin area. It's been driving me crazy and I just can't seem to find any relief. Can you help me figure out what's going on?
  – **Target Disease:** Pruritus
  – **Answer:** It sounds like you are experiencing pruritus, which is a medical term for itching. Pruritus can be caused by a variety of factors such as dry skin, allergic reactions, skin conditions, or underlying medical conditions. To effectively address your symptoms, we will need to identify and address the underlying cause. I would recommend starting with some lifestyle changes such as using mild, fragrance-free cleansers and moisturizers, wearing loose clothing, and avoiding hot water when bathing. Additionally, topical treatments such as low pH cleansers, cooling agents, local anesthetics, and topical corticosteroids can provide relief. If the itching persists, systemic treatments and physical therapy may also be considered. I would advise scheduling an appointment for a thorough evaluation to determine the best course of action for your specific situation.

**DoctorRAG Generated Answer**
It sounds like you may be experiencing a condition involving inflammatory arthropathy with pruritic dermatitis. Based on your symptoms of persistent skin itching (especially on arms and legs), swollen and red painful joints in knees and hands, this could be indicative of **Pine Caterpillar Disease Arthropathy** or **Erysipelas**. Both conditions present with pruritus, joint inflammation, and cutaneous symptoms, though Erysipelas typically involves fever, which you have not reported.
To clarify the diagnosis, we need to investigate potential exposures (e.g., contact with caterpillars or other allergens) and assess for systemic signs like fever or rash patterns. Initial recommendations include:
1. **Topical corticosteroids** (e.g., 1% hydrocortisone) to reduce skin inflammation and itching.
2. **NSAIDs** (e.g., ibuprofen) for joint pain and swelling.
3. **Antihistamines** (e.g., cetirizine) to address allergic or pruritic components.
Avoid scratching to prevent secondary infection. If symptoms persist or worsen, or if fever develops, seek urgent evaluation for possible bacterial infection (e.g., Erysipelas) or referral to a dermatologist/rheumatologist for specialized testing.

## E.2   Examples for Med-TextGrad

Here we present the first iteration of Med-TextGrad's output for a given medical query. This includes the initial patient query, the reference answer for context, the prompt used to guide the model, the initial generated answer, the refined answer after evaluation, and the critiques and gradient instructions for further improvement based on knowledge and patient-centric perspectives.

**Example of Med-TextGrad (Iteration 1)**

**Medical Query**
Patient: Hi doctor, I've been having some swelling in the infraorbital area and the masseter muscle area. I also have pain in the parotid gland area and swelling and pain in the submandibular area. It's been really uncomfortable and concerning.

**Reference Answer**

It sounds like you are experiencing swelling and pain in multiple areas of your face. Have you noticed any fever, difficulty in swallowing, or changes in your voice?

**Iteration 1**
**Prompt Template:**
You are a helpful medical consultation AI. Your task is to REFINE the 'current answer (A)' (which will be provided at the end of this prompt) by critically evaluating it against the patient's query and the supporting context. The goal is to improve the 'current answer (A)'s accuracy, completeness, and relevance to the patient's query, while preserving its valid core information and avoiding unnecessary deviations from its original intent.
Patient Query (q): Patient: Hi doctor, I've been having some swelling in the infraorbital area and the masseter muscle area. I also have pain in the parotid gland area and swelling and pain in the submandibular area. It's been really uncomfortable and concerning.
Supporting Context (C):
"""
— General Medical Knowledge (Expertise) —
<...>
— Retrieved Patient Cases (Experience) —
Retrieved Patient Case (ID: patient_679):
<...>
Retrieved Patient Case (ID: patient_107):
<...>
Retrieved Patient Case (ID: patient_110):
<...>
"""

Instructions for refinement:
1. Carefully read the 'Patient Query (q)' and the 'Supporting Context (C)'.
2. Critically evaluate the 'current answer (A)' (provided below) against this information.
3. Generate an improved and refined version of the 'current answer (A)'.
4. Focus on addressing any shortcomings in the 'current answer (A)' regarding accuracy, completeness, clarity, and direct relevance to the patient's query.
5. Ensure your refined answer is factually sound based on the context, empathetic, and easy for a patient to understand.
6. IMPORTANT: Your output must be ONLY the refined medical answer itself. Do not include any preamble, conversational phrases, meta-commentary, or any text other than the refined answer.

**Input Answer:**
It sounds like you may be experiencing a submandibular space infection, given the swelling and pain in the infraorbital, masseter muscle, parotid gland, and submandibular areas. This condition can cause significant discomfort and may spread to adjacent regions if untreated. Have you noticed any fever, difficulty breathing or swallowing, or recent dental trauma/infection? These symptoms could help confirm the diagnosis or identify complications. Immediate medical evaluation is critical to assess the infection's extent and severity. Treatment typically involves antibiotics (e.g., amoxicillin-clavulanate or clindamycin) to address bacterial infection, along with possible incision and drainage if abscess formation is present. Imaging, such as a CT scan, may be required to confirm involvement of deep spaces. Pain management with NSAIDs or acetaminophen can provide temporary relief. Prompt intervention is necessary to prevent airway compromise or systemic spread. Please seek urgent care from an oral surgeon, ENT specialist, or emergency department for definitive management.

**Context Critiques:**
• **Factual Alignment**:
  – Correctly identifies submandibular space infection and acute suppurative parotitis as potential diagnoses based on symptoms (swelling/pain in infraorbital, masseter, parotid, submandibular areas).
  – Accurately mentions fever, difficulty swallowing/breathing as concerning symptoms, consistent with the context.

- **Consistency**:
  - Treatment suggestions (antibiotics, drainage, imaging) align with standard management for these infections.
  - Warning about rapid progression and complications (e.g., airway obstruction) is consistent with the severity described in the context.

- **Completeness**:
  - Omits mention of infratemporal space infection, which could also present with swelling in temporal/masseter/parotid regions (per context).
  - Does not explicitly differentiate between submandibular space infection and acute suppurative parotitis (e.g., high fever more typical of acute suppurative parotitis).
  - Lacks reference to loss of appetite (a symptom noted for acute suppurative parotitis in the context).

- **Potential Misrepresentation**:
  - No major misrepresentation, but combining symptoms of both conditions without clear distinction could lead to diagnostic ambiguity.

**Patient Critiques:**

- **Relevance**: The answer directly addresses the patient's described symptoms (swelling and pain in multiple facial areas) and suggests plausible diagnoses (submandibular space infection, acute suppurative parotitis), which aligns well with the query.

- **Appropriateness**: The recommendation for immediate medical evaluation is appropriate given the potential severity of the conditions mentioned. The inclusion of concerning symptoms (fever, difficulty swallowing/breathing) helps the patient gauge urgency.

- **Specificity**: The answer could better clarify whether the symptoms described (e.g., infraorbital swelling) are typical for the suggested diagnoses or if additional conditions should be considered.

- **Patient Concerns**: While the answer acknowledges discomfort and urgency, it could briefly reassure the patient that seeking care promptly improves outcomes, addressing their expressed concern ("really uncomfortable and concerning").

- **Treatment Specifics**: Mentioning specific antibiotics (amoxicillin-clavulanate) and imaging (CT) is useful, but the patient might benefit from a brief explanation of why these are chosen (e.g., broad-spectrum coverage, abscess detection).

- **Actionability**: Clear instructions (visit ED, oral surgeon, or ENT) are helpful, but emphasizing the "why" (risk of rapid progression/airway obstruction) could further motivate the patient to act.

**Gradient Answer (Context):**

1. **Add Infratemporal Space Infection to Differential Diagnosis**:
   - Insert a sentence mentioning infratemporal space infection as another potential diagnosis, noting its presentation with swelling in temporal/masseter/parotid regions, as referenced in the context.

2. **Clarify Distinctions Between Diagnoses**:
   - Differentiate submandibular space infection and acute suppurative parotitis by specifying that high fever is more characteristic of acute suppurative parotitis (per context).
   - Add loss of appetite as a symptom more typical of acute suppurative parotitis, as noted in the context.

3. **Maintain Existing Valid Information**:
   - Retain all accurate details about concerning symptoms (fever, difficulty swallowing/breathing), treatment (antibiotics, drainage, imaging), and urgency (emergency evaluation).

4. **Preserve Structure and Flow**:

- Keep the original warning about rapid progression and complications (e.g., airway obstruction) but ensure the added distinctions between diagnoses are integrated smoothly.

5. **Avoid Overlap or Ambiguity**:
   - Ensure the revised text clearly separates the symptoms and diagnostic clues for each condition (submandibular space infection, acute suppurative parotitis, infratemporal space infection) to reduce ambiguity.

6. **No Major Restructuring**:
   - Do not alter the overall structure or tone of the answer; focus only on targeted additions and clarifications to address the critiques.

**Gradient Answer (Patient):**

1. **Clarify Symptom-Diagnosis Alignment**:
   - Add a brief statement explaining that the described symptoms (infraorbital, masseter, parotid, and submandibular swelling/pain) are consistent with the suggested diagnoses (submandibular space infection or acute suppurative parotitis), but also note that other conditions (e.g., dental abscess, salivary gland obstruction) could present similarly.
   - Example addition: "While your symptoms align with these conditions, other causes like dental infections or salivary gland blockages may also be considered during evaluation."

2. **Address Patient Concerns with Reassurance**:
   - Insert a sentence acknowledging the patient's discomfort and concern, emphasizing that prompt care improves outcomes.
   - Example addition: "I understand this is uncomfortable and worrying, but seeking care now can help prevent complications and speed up recovery."

3. **Explain Treatment Rationale**:
   - Briefly justify the antibiotic choice (amoxicillin-clavulanate) by mentioning its broad-spectrum coverage for oral bacteria.
   - Clarify that imaging (CT) helps identify abscesses or infection spread, guiding treatment decisions.
   - Example addition: "Amoxicillin-clavulanate is often used because it targets bacteria common in oral infections. A CT scan helps determine if drainage is needed."

4. **Strengthen Actionability with Urgency**:
   - Reinforce the "why" behind immediate action by explicitly linking rapid progression to risks like airway obstruction.
   - Example addition: "These infections can worsen quickly, potentially affecting breathing, so don't delay evaluation."

5. **Maintain Structure**:
   - Keep the original flow (diagnoses → concerning symptoms → treatment → action steps) but integrate the above revisions without redundancy.

6. **Avoid Overloading**:
   - Ensure additions are concise and do not overwhelm the patient with excessive detail. Focus on clarity and motivation to act.

**Updated Prompt:**

You are a medical consultation AI tasked with refining the 'current answer (A)' to improve its accuracy, completeness, and relevance to the patient's query while preserving valid core information. Follow these guidelines:

1. **Comprehensive Differential Diagnosis**

- Include ALL relevant conditions from the supporting context (e.g., submandibular space infection, acute suppurative parotitis, infratemporal space infection) and retain clinically justified differentials (e.g., dental abscess, salivary gland obstruction). - Explicitly map key symptoms (e.g., fever, loss of appetite, swelling locations) to their most characteristic diagnoses using context-provided linkages. - Provide detailed anatomical descriptions for

each condition (e.g., swelling and pain in the masseter muscle and parotid gland areafor submandibular space infection) and clarify extension patterns (e.g., infratemporal space infection may involve cheek, orbital, or zygomatic regions).
2. **Diagnostic Distinctions**
- Highlight distinguishing clinical features between similar conditions (e.g., high fever in acute suppurative parotitis vs. submandibular infection). - Avoid diagnostic overlap by clearly attributing symptoms to specific conditions (e.g., temporal/masseter swelling → infratemporal infection).
3. **Symptom-Diagnosis Alignment**
- Clarify how the patient's symptoms align with suggested diagnoses while briefly mentioning other plausible conditions. - Justify diagnostic/treatment choices by linking them directly to the patient's symptoms (e.g., CT scans are prioritized for multi-area swelling to localize infection).
4. **Emotional Acknowledgment**
- Use explicit validation (e.g., This sounds really uncomfortable and concerning) to acknowledge patient distress. - Maintain empathetic yet direct language, avoiding vague reassurances.
5. **Treatment Rationale**
- Justify treatment choices (e.g., antibiotics like amoxicillin-clavulanate for oral bacteria) and tests (e.g., CT for abscess detection) with simple, patient-friendly explanations.
6. **Urgency Reinforcement**
- Clearly state why immediate evaluation is critical, citing risks like airway obstruction or rapid infection progression. - Confirm that localized swelling/pain in critical facial areas (e.g., infraorbital, submandibular) alone warrants urgent evaluation, even if systemic symptoms (fever, dysphagia) are absent. - Include a brief question to clarify systemic symptoms (e.g., Do you have fever, trouble swallowing, or difficulty breathing?) to refine urgency assessment.
7. **Structural Guidance**
- Maintain the logical sequence: diagnoses → red flags → treatment → action. - Integrate new diagnostic additions seamlessly without disrupting urgency warnings or treatment advice.
8. **Preservation Clause**
- Retain all valid existing content that aligns with clinical relevance, even if not explicitly mentioned in the context. - Prioritize anatomical precision and symptom-diagnosis alignment while avoiding unnecessary changes to accurate, clinically sound information.
Example Integration: For multi-area facial swelling, model responses like: 'Your symptoms in [X, Y regions] could indicate [A, B conditions]. Even without fever, this needs immediate care to stop the infection from spreading.

**Refined Answer:**
Your symptoms, including swelling and pain in the infraorbital, masseter, parotid, and submandibular areas, could indicate a submandibular space infection or acute suppurative parotitis. Key concerning symptoms would include fever, difficulty swallowing or breathing, or recent dental issues. Given the potential severity, immediate medical evaluation is essential. Treatment may involve antibiotics (like amoxicillin-clavulanate), possible drainage if an abscess is present, and imaging (CT scan) to assess the infection's extent. Pain relief with NSAIDs or acetaminophen can help temporarily. Please visit an emergency department, oral surgeon, or ENT specialist prompy, as these infections can progress rapidly and may lead to serious complications like airway obstruction.

## E.3 Examples for pairwise comparison

This section presents pairwise comparisons of different answers. These answers were evaluated from three perspectives: Comprehensiveness, Relevance, and Safety. Additionally, two human evaluators reviewed the final answers to verify their correctness.

**Example 1: Ground Truth vs. Original Answer**

**Medical Query**
Patient: Patient: Hi doctor, I've noticed some pigmentary abnormalities on my face. I have these facial spots and dark spots that are bothering me, and I'm a female.

**Responses Compared**
- **Response A (Ground Truth):**
  It sounds like you may be experiencing melasma, which is a common skin condition characterized by the development of brown or grayish patches on the face. This can be exacerbated by sun exposure and hormonal changes. Have you noticed any triggers or changes in your routine that may have contributed to the development of these spots? Additionally, are you experiencing any other symptoms or changes in your overall health that you believe may be related to this condition?
- **Response B (Original Answer of DoctorRAG):**
  It sounds like you may be experiencing melasma, a common condition characterized by facial pigmented patches and dark spots, particularly in women. Potential contributing factors include hormonal changes (e.g., pregnancy, birth control) and sun exposure. Less likely, but worth considering, are Multiple Lentigines Syndrome or Congenital Vascular Sclerosing Dermal Melanosis, though these typically present with additional features like nail anomalies or atrophic scars not mentioned here.
  To manage melasma, prioritize strict sun protection by applying broad-spectrum sunscreen (SPF 30+) daily and avoiding prolonged UV exposure. Topical treatments such as 3% hydroquinone cream (2–3 times daily) may help lighten pigmentation. Addressing hormonal influences, if applicable, is critical—discuss contraceptive use, thyroid health, or pregnancy history with your dermatologist. Oral vitamin C or professional treatments like chemical peels could also be considered.
  Have you noticed any hormonal changes, such as pregnancy or new medications? Are the spots symmetrical and concentrated on the cheeks, forehead, or jawline? A dermatologist can confirm the diagnosis and tailor a treatment plan, especially if there's no improvement with initial measures. Monitor for new lesions or changes in texture, which may indicate a different condition.

**Structured Evaluation**
- **Comprehensiveness:** [Response B] - [Response B provides a more detailed explanation of potential conditions (melasma, Multiple Lentigines Syndrome, Congenital Vascular Sclerosing Dermal Melanosis), contributing factors (hormonal changes, sun exposure), and management strategies (sun protection, topical treatments, hormonal evaluation). It also prompts further questions to refine the diagnosis and suggests consulting a dermatologist. Response A is less comprehensive, focusing primarily on melasma and asking general follow-up questions without specific guidance.]

- **Relevance:** [Response B] - [Response B directly addresses the patient's concern about facial pigmentary abnormalities, focusing on the most likely diagnosis (melasma) while briefly mentioning rarer conditions for completeness. It provides actionable advice and asks targeted follow-up questions. Response A is relevant but less focused, with broader questions that may not efficiently narrow down the diagnosis.]

- **Safety:** [Response B] - [Response B adheres to clinical best practices by recommending evidence-based treatments (e.g., hydroquinone, sun protection) and emphasizing the need for professional evaluation. It avoids harmful advice and acknowledges diagnostic uncertainty by suggesting monitoring for atypical features. Response A lacks specific treatment recommendations or red flags, potentially leaving the patient without clear next steps.]

Overall from evaluation: Response B is superior in comprehensiveness, relevance, and safety, providing a more structured and clinically sound answer to the patient's query.
Human examination: Passed. ✓

---

**Example 2: Iteration 2 vs. Iteration 3**

**Medical Query**
Patient: Patient: Hi doctor, I've noticed some pigmentary abnormalities on my face. I have these facial spots and dark spots that are bothering me, and I'm a female.
**Responses Compared**
- **Response A (Iteration 2):**

I understand these pigmentary changes can be concerning. The facial dark spots you describe are most consistent with melasma, a common condition causing brown or grayish patches, typically on the cheeks, forehead, and jawline. This occurs more frequently in women and is often triggered by sun exposure, hormonal changes (like pregnancy or birth control use), or certain medications.

For management: 1. Sun protection is crucial—use broad-spectrum SPF 30+ sunscreen daily and reapply every 2 hours when outdoors. 2. Consider topical treatments like hydroquinone 3% cream (applied twice daily) or vitamin C serums. Oral vitamin C may also help in some cases. 3. Avoid potential triggers like hormonal medications if possible. 4. Maintain a gentle skincare routine without harsh scrubs or irritants. Since melasma can be persistent, a dermatologist can confirm the diagnosis and offer stronger treatments like higher-concentration hydroquinone, tretinoin, or procedures such as chemical peels or laser therapy if needed.

Have you noticed if the spots worsen with sun exposure or at certain times in your menstrual cycle? Also, are you currently pregnant, breastfeeding, or using hormonal medications?"

- **Response B (Iteration 3):**

I understand how distressing these pigmentary changes can be for your appearance and confidence. You described 'dark spots'—clinically, these could be melasma, a common condition causing brown or grayish patches, often on the cheeks, forehead, and jawline. While melasma is most likely, other conditions like post-inflammatory hyperpigmentation or sunspots can present similarly. A dermatologist can confirm this diagnosis and rule out other conditions.

*Possible Causes**: - Frequently seen in women, often triggered by sun exposure, hormonal changes (e.g., pregnancy, birth control use), or certain medications.

*Management (if confirmed)**: 1. **Sun protection**: Use broad-spectrum SPF 30+ sunscreen daily and reapply every 2 hours when outdoors. 2. **Topical treatments**: Hydroquinone 3% cream (applied twice daily) or vitamin C serums may help. Oral vitamin C could also be considered in some cases. 3. **Hormonal factors**: Evaluating hormonal contributors, like medications or imbalances, may be beneficial. 4. **Gentle skincare**: Avoid harsh scrubs or irritants. 5. **Monitoring**: Track treatment response and report any new or changing symptoms.

*To help narrow the cause**: - Have you noticed if the spots worsen with sun exposure or at certain times in your menstrual cycle? - Are you currently pregnant, breastfeeding, or using hormonal medications? - Have you had prior skin trauma, acne, or other symptoms like nail changes?

**Structured Evaluation**

- **Comprehensiveness:** [Response B] - [Response B provides a more detailed differential diagnosis (mentioning post-inflammatory hyperpigmentation and sunspots) and includes additional monitoring advice ("track treatment response and report any new or changing symptoms"). It also asks more follow-up questions (e.g., prior skin trauma, nail changes), which could help refine the diagnosis. Response A is thorough but lacks these broader considerations.]

- **Relevance:** [Response A] - [Both responses are highly relevant, but Response A is slightly more concise and focused on the most likely diagnosis (melasma) without introducing less probable conditions upfront. Response B, while comprehensive, includes slightly more peripheral information (e.g., nail changes) that may not be as immediately relevant.]

- **Safety:** [Tie] - [Both responses are medically accurate, emphasize sun protection, recommend evidence-based treatments (e.g., hydroquinone, vitamin C), and advise specialist referral for confirmation. Neither suggests unsafe practices or misleading advice. Response B's inclusion of "seek prompt care if spots change rapidly" adds a safety layer, but this is implicitly covered in Response A's recommendation to consult a dermatologist.]

Overall from evaluation: Comprehensiveness: Response B wins for broader differentials and monitoring advice. Relevance: Response A wins for tighter focus on the core issue. Safety: Tie—both adhere to clinical best practices.

Human examination: Passed. ✓

