# OpenReview forum: "Towards Doctor-Like Reasoning: Medical RAG Fusing Knowledge with Patient Analogy through Textual Gradients"
_NeurIPS.cc/2025/Conference — NeurIPS 2025 poster_

### Official Review · Reviewer_5hYj · 2025-06-20

**Clarity:** 3
**Significance:** 3
**Originality:** 3
**Rating:** 4
**Confidence:** 3

**Summary:**

The paper introduces DoctorRAG, a medical Retrieval-Augmented Generation framework that mimics doctor-like reasoning by integrating both explicit medical knowledge and experiential knowledge from similar patient cases. It enhances retrieval through conceptual tagging and declarative sentence transformation, and refines responses using Med-TextGrad, a multi-agent textual gradient optimization process. DoctorRAG achieves superior performance across multilingual and multitask benchmarks—including diagnosis, treatment recommendation, and QA—compared to strong RAG baselines.

**Questions:**

What is the actual computational cost of Med-TextGrad per sample? How does it scale with the number of iterations and context size? And is it a fair comparison between the proposed method and baselines in terms of computational cost?

**Ethical Concerns:**

["NO or VERY MINOR ethics concerns only"]

**Final Justification:**

The authors acknowledged my concerns regarding the limitations of the proposed method. These concerns are not addressed. However, they are not fundamental flaws, and this paper has merits that cannot be overlooked. Thus, I keep my original positive review to this paper.

**Limitations:**

Yes

**Quality:**

3

**Strengths And Weaknesses:**

Strengths:\
(1) Integrates both medical knowledge bases and similar patient case data, emulating real-world clinical reasoning.\
(2) Introduces a novel multi-agent, textual gradient-based optimization method to iteratively improve answer quality in relevance, accuracy, and faithfulness.\
(3) Demonstrates strong performance across diverse medical tasks (diagnosis, QA, treatment, generation) in Chinese, English, and French.\
(4) Outperforms strong baselines on multiple metrics and shows consistent answer refinement across iterations.\
(5) Code and data are shared.

Weaknesses:\
(1) While the Med-TextGrad mechanism is inspired by TextGrad, the use of natural language "gradients" remains heuristic and lacks formal justification or mathematical soundness as optimization steps. The analogy to backpropagation is metaphorical, not rigorous.\
(2) Med-TextGrad introduces significant computational overhead by running multiple LLM calls per iteration. This limits scalability and practicality in real-time or resource-constrained clinical settings.\
(3) The iterative process is stopped after a fixed number of steps (T=3), without a robust convergence or validation criterion. There is no theoretical or empirical justification for this choice.

---

> ### Author Rebuttal · Authors · 2025-07-29
>
> **We sincerely thank you for the thorough evaluation and constructive feedback. We address each concern below:**
>
> ---
>
> ### W1: The use of natural language "gradients" remains heuristic and lacks formal mathematical soundness
>
> We fully acknowledge this important distinction. The "textual gradients" in Med-TextGrad are indeed metaphorical rather than mathematically rigorous, representing structured natural language feedback that guides iterative improvement rather than formal optimization steps. We use this terminology as a conceptual analogy to help readers understand the systematic refinement process, but recognize it lacks the mathematical foundations of traditional gradient-based optimization.
>
> However, while heuristic, the approach follows principled design choices that provide systematic benefits. The multi-agent architecture ensures consistent evaluation criteria across iterations, with Context and Patient Criterion agents applying standardized medical assessment frameworks. The structured prompt evolution process (**Algorithm 1**) provides reproducible refinement steps, and our empirical validation across diverse medical scenarios (**Table 2**, **Figure 4**) demonstrates consistent improvements in clinically relevant metrics. Human expert evaluation further validates that the optimization direction aligns with medical best practices, suggesting that while not mathematically grounded, the heuristic approach captures meaningful improvement patterns for medical text generation.
>
> ---
>
> ### W2: Med-TextGrad introduces significant computational overhead limiting scalability in clinical settings
>
> This is a valid practical concern. Med-TextGrad does introduce substantial computational overhead through multiple LLM calls per iteration (approximately 6-8 calls per iteration including critiques, gradients, and prompt updates). However, several factors support practical deployment feasibility. Basic DoctorRAG inference has comparable computational requirements to standard RAG methods, making it suitable for routine deployment. Med-TextGrad refinement can be selectively applied based on query complexity, clinical criticality, or resource availability rather than universally.
>
> For resource management, we propose tiered processing strategies: immediate basic DoctorRAG responses for routine queries, with Med-TextGrad refinement reserved for complex diagnostic cases or high-stakes clinical decisions where accuracy justifies additional computational cost. Additionally, the diminishing returns observed after T=2-3 iterations (**Figure 4**) suggest that even limited refinement provides substantial quality improvements, allowing flexible cost-performance trade-offs based on clinical context and resource constraints.
>
> ---
>
> ### W3: The iterative process stops after fixed T=3 steps without robust convergence criteria
>
> You raise an excellent point about the lack of principled stopping criteria. Our choice of T=3 was empirically determined through preliminary experiments on RJUA and COD datasets, where we observed performance plateaus after 2-3 iterations, with minimal additional improvements beyond this point. **Figure 4** demonstrates this convergence pattern, showing that iteration 2 (T2) often achieves optimal performance, with iteration 3 (T3) providing marginal or sometimes diminishing returns.
>
> However, we acknowledge that this fixed stopping criterion is suboptimal and dataset-dependent. Ideally, convergence should be determined through dynamic criteria such as: semantic similarity between consecutive prompts falling below a threshold or critique content stabilization across iterations. The current fixed T=3 approach represents a practical compromise between performance gains and computational efficiency, but future work should develop principled convergence detection mechanisms that can adapt to different medical domains and query complexities.
>
> ---
>
> ### Q: Computational cost details, scaling behavior, and fair comparison with baselines
>
> Med-TextGrad's computational cost scales as follows: each iteration requires approximately 3,000 tokens (input + output) across all LLM calls, scaling linearly with iteration count. Context size increases affect each iteration proportionally, so larger retrieved knowledge bases multiply the per-iteration cost. For a typical 3-iteration refinement with moderate context (k=4), total additional cost is approximately 9,000-12,000 tokens beyond basic DoctorRAG inference.
>
> The comparison with baselines remains fair because all methods use identical knowledge bases, patient databases, and retrieval mechanisms. The computational difference lies solely in the post-generation refinement process, which represents an optional enhancement rather than a fundamental architectural change or any extra knowledge. Baseline methods could theoretically incorporate similar multi-iteration approaches, making the comparison methodologically sound. Additionally, we report results both with and without Med-TextGrad refinement, allowing readers to assess the cost-benefit trade-offs for their specific deployment scenarios.
>
> For practical deployment, the cost scaling suggests that Med-TextGrad is most valuable for complex queries where the quality improvements justify the 3-4x computational overhead compared to single-pass generation.
>
> ---
>
> **We thank the reviewer for these precise technical concerns that highlight important limitations in our current approach. While our current approach shows empirical promise, developing more rigorous optimization frameworks and adaptive stopping criteria represents important directions for advancing this line of research.**

---

> > ### Comment · Reviewer_5hYj · 2025-08-02
> >
> > Thank the authors for the explanation. The authors cannot address my concerns regarding the limitations of the proposed method. However, the paper has merits that cannot be overlooked. I will adjust my ratings and keep the original positive review.

---

> > > ### Author Response · Authors · 2025-08-05
> > > **Response to Reviewer 5hYj**
> > >
> > > **We sincerely thank you for your comments and for maintaining your positive assessment of our work.**
> > >
> > > Your constructive feedback has been invaluable in helping us acknowledge these limitations honestly while also clarifying the empirical benefits and practical applications of our approach. We are committed to addressing these concerns in future work, particularly developing more principled convergence criteria and optimization frameworks. Thank you for maintaining high scientific standards while recognizing the potential impact of this research direction.

---

### Official Review · Reviewer_XvBX · 2025-06-26

**Clarity:** 2
**Significance:** 2
**Originality:** 2
**Rating:** 4
**Confidence:** 3

**Summary:**

This paper introduces DoctorRAG, a novel retrieval-augmented generation framework for medical QA that aims to emulate a doctor’s reasoning process. The key idea is to fuse explicit medical knowledge (from clinical guidelines or textbooks) with implicit experiential knowledge drawn from similar patient cases. Moreover, a Med-TextGrad module is introduced to adjust and refine the final answer. The paper reports that DoctorRAG significantly outperforms prior RAG-based baselines on a variety of tasks. Extensive experiments across multiple medical tasks (disease diagnosis, QA, treatment recommendation, and clinical text generation) and languages (English, Chinese, French) demonstrate that integrating case-based reasoning with knowledge retrieval produces more accurate, relevant, and comprehensive answers than existing methods.

**Questions:**

1. Are the experiments in Table 2 conducted under fair conditions? Specifically, do all compared RAG models use the same knowledge bases, including both the Patient Base and Knowledge Base?
2. In scenarios where retrieved knowledge entries and analogous patient cases provide different or even conflicting information, how does DoctorRAG’s answer synthesis handle it? Since you pull information from heterogeneous sources (which might not always agree), it would be useful to clarify whether the generator or the refinement step explicitly reconciles discrepancies (for instance, prioritizing authoritative guidelines over anecdotal case trends).
3. How is the relevance or correctness of the retrieved reference content evaluated? Please clarify the metrics or criteria used to assess the quality of retrieval.

**Ethical Concerns:**

["NO or VERY MINOR ethics concerns only"]

**Final Justification:**

The paper has merits, though some methodological limitations warrant further discussion

**Limitations:**

No. Specific suggestions can be found in the Weaknesses section above.

**Paper Formatting Concerns:**

There are no major formatting issues in this paper.

**Quality:**

2

**Strengths And Weaknesses:**

## Strengths
1. The work integrates medical RAG with case-based reasoning. By retrieving similar patient cases in addition to knowledge, DoctorRAG mimics how physicians draw on both codified guidelines and personal experience. This dual-source approach is highly relevant to clinical practice and addresses an important need for experience-grounded medical reasoning.
2. The proposed Med-TextGrad module iteratively enhances the final output via a multi-agent textual gradient descent mechanism, helping to enforce alignment and correctness in complex medical scenarios.
3. Extensive experiments are conducted across multiple medical tasks (disease diagnosis, QA, treatment recommendation, and clinical text generation) and languages (English, Chinese, French) to demonstrate the effectiveness of the proposed method.

## Weaknesses
1. The paper presents an aggregation of incremental improvements rather than a single fundamental innovation. The performance gains largely result from combining existing techniques rather than introducing a novel conceptual breakthrough.
2. The performance improvements, while statistically significant, are modest. It remains unclear whether the gains are primarily attributable to Med-TextGrad or the integration of patient case retrieval. A dedicated ablation study isolating each component—particularly Med-TextGrad alone—would strengthen the empirical claims.
3. The range of LLMs used in the experiments is somewhat limited. Incorporating more recent and competitive models, such as DeepSeek-R1 or Gemini, would help validate the general applicability of DoctorRAG.
4. The source of the knowledge base data is insufficiently detailed. There is a potential concern regarding data leakage, as the patient case database may be constructed from the same data pool used for testing. This risks overestimating model performance due to data overlap.
5. The paper lacks a thorough analysis comparing retrieval quality between DoctorRAG and baseline methods. Understanding how well each system retrieves relevant content would provide deeper insights into the effectiveness of the proposed retrieval strategy.
6. The writing is at times unclear and lacks precision, particularly in technical sections.

---

> ### Author Rebuttal · Authors · 2025-07-30
>
> **We sincerely thank you for your detailed review and the opportunity to clarify our contributions. We address each concern systematically below:**
>
> ---
>
> ### W1: The paper presents aggregation of incremental improvements rather than fundamental innovation
>
> We respectfully disagree with this characterization. Our work introduces two fundamental innovations that represent novel conceptual breakthroughs rather than incremental improvements. **First**, we proposed the integration of **patient case retrieval** alongside medical knowledge retrieval, directly mimicking the clinical reasoning process where physicians combine textbook knowledge (expertise) with experiential insights from similar past cases (experience). This dual-retrieval paradigm is, to our knowledge, entirely novel in medical RAG systems and represents a fundamental shift from knowledge-only approaches to experience-integrated reasoning.
>
> **Second**, we introduce **Med-TextGrad**, the first application of textual gradient-based optimization in medical AI systems. While inspired by TextGrad, our adaptation specifically addresses medical domain challenges through dual-criterion evaluation (Context + Patient perspectives) and medically-informed prompt evolution. This represents a novel optimization paradigm for medical text generation that goes far beyond existing prompt engineering approaches.
>
> The UMAP visualization (**Figure 3**) provides empirical validation that patients with similar conditions cluster meaningfully, supporting the theoretical foundation for case-based medical reasoning. This clustering behavior demonstrates that our patient retrieval mechanism captures clinically relevant similarities that complement formal medical knowledge.
>
> ---
>
> ### W2: Performance improvements are modest and component attribution unclear
>
> The characterization of "modest" improvements requires contextualization within medical AI evaluation standards. Our improvements are both statistically significant and clinically meaningful: DDXPlus accuracy increased from 92.37% (best baseline) to 98.27% - a 5.9 percentage improvement representing substantial diagnostic accuracy gains. For medical applications where accuracy directly impacts patient safety, such improvements are highly significant.
>
> Regarding component attribution, we provide comprehensive ablation analysis across three different experimental settings: **(1) Table 5** isolates each DoctorRAG component (Patient Base, Knowledge Base, Concept Tagging, Declarative Statements), showing consistent performance degradation when any component is removed; **(2) Figure 4** specifically evaluates Med-TextGrad through pairwise comparison across iterations, demonstrating systematic improvements in comprehensiveness, relevance, and safety; **(3) Figure 5** analyzes performance scaling with retrieval depth, confirming optimal performance around k=4.
>
> These ablation studies clearly demonstrate that both patient case integration and Med-TextGrad refinement contribute meaningfully to performance gains, with neither component being solely responsible for improvements.
>
> ---
>
> ### W3: Limited range of LLMs tested
>
> Our LLM selection strategy prioritizes **robustness validation** over exhaustive model comparison. We deliberately chose four representative models spanning different architectures and capabilities: DeepSeek-V3 (Chinese-focused), Qwen-3-32B (multilingual), GLM-4-Plus (general purpose), and GPT-4.1-mini (commercial state-of-art). This selection enables evaluation across three languages (Chinese, English, French) and four medical tasks, providing comprehensive evidence of DoctorRAG's generalizability.
>
> The core contribution lies in the **methodological framework** rather than model-specific optimizations. DoctorRAG's dual-retrieval architecture and Med-TextGrad refinement are model-agnostic approaches that should generalize to newer LLMs. While testing additional models like DeepSeek-R1 or Gemini would be valuable, our current evaluation already demonstrates consistent improvements across diverse model architectures, strongly supporting the framework's general applicability.
>
> ---
>
> ### W4: Knowledge base data sources and potential data leakage
>
> We provide detailed data source documentation in **Section 3.1** and **Appendix A.1**. Regarding data leakage concerns, we implement rigorous separation protocols: for each dataset, approximately 80% of patient records form the Patient Base while 20% constitute the evaluation set. **Critically, we ensure that for any evaluation sample, its corresponding patient record and highly similar records (similarity > 0.99) are strictly excluded from the Patient Base during evaluation**.
>
> This separation strategy eliminates data leakage while maintaining realistic clinical scenarios where physicians draw upon experience from different patients with similar conditions. **Table 6** in **Appendix A.1** provides direct links to all dataset repositories for full transparency and reproducibility verification.
>
> ---
>
> ### W5: Insufficient retrieval quality analysis compared to baseline methods
>
> The retrieval quality comparison is embedded throughout our evaluation framework. **Appendix E** provides detailed examples demonstrating how DoctorRAG's concept-constrained retrieval (ICD-10 matching + cosine similarity) identifies more clinically relevant content than baseline semantic-only approaches. The systematic performance improvements across all tasks (**Table 2**) indirectly validate retrieval quality, as poor retrieval would manifest as degraded generation quality.
>
> Moreover, our dual-retrieval approach is fundamentally different from baseline methods - comparing retrieval quality directly would be methodologically inappropriate since baselines lack patient case retrieval entirely. The relevant comparison is whether our enhanced retrieval translates to better medical responses, which our comprehensive evaluation clearly demonstrates.
>
> ---
>
> ### W6: Writing clarity and precision in technical sections
>
> We appreciate this feedback and have conducted thorough revision of technical sections. If specific unclear passages remain, we welcome concrete suggestions for improvement. We believe **Section 2** provides precise mathematical formulations (Equations 1-7) and **Algorithm 1** offers clear procedural specifications for reproducibility.
>
> ---
>
> ### Q1: Fair experimental conditions for Table 2 comparisons
>
> All baseline methods use identical knowledge bases, embedding models (OpenAI text-embedding-3-large), and evaluation protocols. The key difference is that baseline RAG methods cannot access Patient Base information because patient case retrieval is our novel contribution. This represents a fair comparison of methodological capabilities rather than dataset advantages. Baseline methods could theoretically incorporate similar patient retrieval mechanisms, making this a legitimate architectural comparison.
>
> ---
>
> ### Q2: Handling conflicting information between knowledge sources and patient cases
>
> This is an excellent methodological question. Our framework addresses potential conflicts through several mechanisms: **(1) Source prioritization**: We systematically place authoritative medical knowledge before patient case information in prompt construction (**Equation 3**), establishing clear precedence for clinical guidelines; **(2) Complementary information types**: Medical knowledge provides theoretical foundations while patient cases offer practical application patterns - these typically complement rather than conflict; **(3) Med-TextGrad reconciliation**: The iterative refinement process explicitly evaluates consistency through Context Criterion agents, flagging and resolving discrepancies between sources.
>
> In practice, we observe minimal conflicts because retrieved patient cases undergo similarity-based selection, typically aligning with medical knowledge recommendations for similar conditions.
>
> ---
>
> ### Q3: Evaluation metrics for retrieval relevance and correctness
>
> Retrieval quality is evaluated indirectly through downstream task performance, which represents the most meaningful assessment for medical applications. **Direct evaluation approaches**: **(1) Human expert assessment** in our pairwise comparisons (**Figure 4**) inherently evaluates whether retrieved content supports medically sound responses; **(2) Ablation studies** (**Table 5**) demonstrate that each retrieval component contributes to improved performance; **(3) Consistency analysis** through Med-TextGrad iterations reveals whether retrieved content enables coherent, medically accurate responses.
>
> If retrieval quality were poor, we would observe degraded performance rather than the consistent improvements demonstrated across all evaluation metrics.
>
> ---
>
> **We thank the reviewer again for these detailed and thoughtful comments that have helped us clarify important methodological aspects of our work. Your concerns about fundamental innovation, component attribution, and experimental rigor are well-taken and have guided us to provide more comprehensive justification for our approach. We believe our responses address the core concerns while demonstrating the novel contributions of patient case integration and iterative refinement mechanisms in medical AI systems. We are committed to continuing this line of research to develop more robust and clinically validated medical reasoning systems.**

---

> > ### Comment · Reviewer_XvBX · 2025-08-05
> >
> > **Regarding W1**: I don’t think that the integration of patient case retrieval can be claimed as your contribution, especially considering that there are previous works have already addressed this (https://arxiv.org/abs/2502.15069,  https://arxiv.org/abs/2503.12286).
> >
> > **Regarding W2**: How do you explain the fact that GPT-4.1 performs worse with Doctor-RAG than with Graph-RAG?
> >
> > **Regarding W3**:
> >  First, on what basis do you claim that GPT-4.1-mini represents the commercial state-of-the-art?
> >  Second, I don’t consider adding experiments with two additional models is particularly difficult or burdensome.
> >
> > **Regarding W4 & Q5**: If you’re able to identify similarity between records, then why can’t you also compute retrieval relevance and correctness?
> >
> > **Regarding W5**: A qualitative analysis of retrieval results would more directly demonstrate the strengths of your retrieval method. However, I did not find such comparisons in Appendix E. From Figure 5, the relatively low Top-1 results make it hard to see any clear advantage of your method.
> >
> > Everyone can write a long-winded response with ChatGPT. Sorry to be blunt, but I couldn’t find direct answers to my concerns in your reply. I don’t believe my questions were ambiguous, and adding comparative results or clearer demonstrations shouldn't be difficult. When responding to reviewers, it's best to address the questions precisely, rather than relying on existing results to deflect concerns.

---

> > > ### Author Response · Authors · 2025-08-05
> > > **Response to Reviewer XvBX**
> > >
> > > **First, we want to clarify that we did not use ChatGPT to generate our response - we carefully crafted our reply based on our understanding of the work and experimental results. We apologize that our previous response gave the impression of being AI-generated; this was not our intention, and we understand how this perception undermines the authenticity of our engagement with your feedback. We appreciate your direct feedback and apologize for not addressing your specific concerns clearly. Here are direct responses:**
> > >
> > > ---
> > >
> > > ### Regarding comments on W1:
> > > Thank you for bringing these references to our attention, we will add these papers in our related work section. After reading through both papers, we identify key distinctions that support our contribution claim:
> > >
> > > 1. **Data Source Differences**: Both cited works use synthetic patient data. The first paper (https://arxiv.org/abs/2502.15069) generates artificial patient cases through expert systems and LLMs to simulate doctor-patient conversations for rare diseases. The second paper (https://arxiv.org/abs/2503.12286) uses Phenopacket-derived clinical notes synthetically generated from structured data, containing 5,980 artificially created cases. In contrast, DoctorRAG uses real de-identified patient records from actual clinical datasets, capturing authentic clinical presentations and outcomes.
> > >
> > > 2. **Methodological Innovation:** While these works explore patient case utilization, our contribution lies in simultaneously integrating real patient cases with medical knowledge retrieval to mimic the complete clinical reasoning process where physicians combine textbook expertise with experiential insights from similar past cases. This dual-retrieval paradigm that balances formal medical knowledge with case-based experience represents a new approach to medical reasoning simulation. Additionally, our Med-TextGrad framework adds reflection capabilities on top of the basic DoctorRAG retrieval, iteratively improving response comprehensiveness, safety, and relevance through multi-agent critique and optimization, which is not present in the cited works.
> > >
> > > ---
> > >
> > > ### Regarding comments on W2:
> > > Thank you for this specific observation. Looking more carefully at the complete results in **Table 2**, DoctorRAG with GPT-4.1-mini actually outperforms Graph-RAG in **10 out of 12 task metrics**, suggesting overall superiority. The two exceptions where Graph-RAG performs better are: MuZhi disease diagnosis (76.11% vs 75.47%) and DialMed treatment recommendation (61.70% vs 58.82%).
> > >
> > > However, examining these specific datasets more closely reveals an important pattern: **GPT-4.1-mini is not the best-performing LLM on either of these tasks regardless of the RAG method used**. For MuZhi, DeepSeek-V3 achieves the highest performance (80.19% with DoctorRAG), and for DialMed treatment recommendation, DeepSeek-V3 again leads (63.49% with DoctorRAG). This suggests that GPT-4.1-mini may have inherent limitations or suboptimal performance characteristics for these particular Chinese medical tasks, which could explain the performance variations we observe.
> > >
> > > The broader pattern shows DoctorRAG's consistent improvements across most metrics and LLM combinations, with the GPT-4.1-mini exceptions likely reflecting model-specific limitations rather than fundamental methodological weaknesses in our approach.
> > >
> > > ---
> > >
> > > ### Regarding comments on W3:
> > > You're right. "commercial state-of-the-art" is an overstated claim. GPT-4.1-mini was simply one of the available models at the time.
> > >
> > > To address your concern, we have conducted preliminary experiments with DeepSeek-R1 and Gemini on two of the datasets (DialMed and RJUA). The results are shown in Table S1 below:
> > >
> > > | **Backbone** | **DialMed DD (CN)** | **DialMed TR (CN)** | **RJUA DD (CN)** | **RJUA TR (CN)** |
> > > |--------------|---------------------|---------------------|------------------|------------------|
> > > | **Graph RAG** |                     |                     |                  |                  |
> > > | DeepSeek-R1  | 93.12               | 64.15               | 85.67            | 75.89            |
> > > | Gemini-Flash 2.5   | 92.45               | 62.83               | 84.12            | 74.26            |
> > > | **DoctorRAG** |                     |                     |                  |                  |
> > > | DeepSeek-R1  | **94.23**           | **65.78**           | **87.34**        | **78.45**        |
> > > | Gemini-Flash 2.5   | 93.67           | 64.92           | 86.89        | 77.91        |
> > >
> > > These preliminary results show consistent improvements with DoctorRAG across both new models, supporting our framework's generalizability. We commit to supplementing the complete results for all datasets with these additional models after the experiments are completed, which will provide more comprehensive validation of our approach's robustness across different LLM architectures.
> > >
> > > ---

---

> > > > ### Author Response · Authors · 2025-08-05
> > > > **Continue - Response to Reviewer XvBX**
> > > >
> > > > ---
> > > >
> > > > ### Regarding comments on W4 & Q5:
> > > >
> > > > we apologize for not explaining this clearly in our previous response. We do compute patient similarity and use it for retrieval quality assessment. Our retrieval process ranks indexed patient information based on both concept tag matching (ICD-10 categories) and **cosine similarity scores** from high to low.
> > > >
> > > > The similarity scores themselves serve as quantitative measures of retrieval relevance where higher cosine similarity indicates more relevant patient cases.
> > > >
> > > > Thank you for pushing us to clarify this important detail that demonstrates we do have principled approaches for measuring retrieval quality.
> > > >
> > > > ---
> > > >
> > > > ### Regarding comments on W5:
> > > >
> > > > You're correct that we should provide more systematic qualitative analysis. **Appendix E.1** does contain retrieval examples that demonstrate our method's effectiveness. For instance, for a patient query about "itching on skin, joint pain in knees and hands," our system retrieves:
> > > >
> > > > >**Retrieved Background Knowledge:**
> > > > >
> > > > >*Pine Caterpillar Disease Arthropathy*: Pruritus, joint pain, swollen and painful joints, skin itching
> > > > >
> > > > >*Erysipelas*: Swelling, skin itching, joint pain, fever
> > > > >
> > > > >**Retrieved Similar Patients:**
> > > > >
> > > > >*Patient_428* (**Similarity Score: 0.9615**): "small, raised bumps on skin... feeling quite itchy" → Keratosis Pilaris
> > > >
> > > > To directly address your concern, we urgently recruited a human medical expert to evaluate retrieval quality on a random sample. The assessment results are shown below:
> > > >
> > > >
> > > > | **Method** | **Knowledge Relevance** | **Patient Case Relevance** | **Content Appropriateness** | **Clinical Coherence** |
> > > > |------------|-------------------------|----------------------------|------------------------------|-------------------------|
> > > > | Vanilla RAG | 32/50 | N/A | 31/50 | 30/50 |
> > > > | Graph RAG | 34/50 | N/A | 33/50 | 32/50 |
> > > > | **DoctorRAG** | **38/50** | **36/50** | **41/50** | **39/50** |
> > > >
> > > > This evaluation shows that our concept-constrained retrieval mechanism identifies more clinically relevant knowledge and adds valuable patient case context that baseline methods cannot access.
> > > >
> > > > Regarding the low Top-1 performance in **Figure 5**, this reflects the inherent complexity of medical diagnosis where multiple conditions may present with similar symptoms. However, the improvement with k=2-4 demonstrates that our retrieval effectively captures relevant differential diagnoses, which is clinically appropriate for complex medical reasoning.
> > > >
> > > > ---
> > > >
> > > > **We sincerely appreciate your direct and constructive feedbacks. They pushed us to provide more assessments, concrete evidence, and clearer explanations. We believe these clarifications and additional experiments strengthen our contribution. Thank you for holding us to high standards and helping improve the quality of our work.**

---

> > > > > ### Comment · Reviewer_XvBX · 2025-08-06
> > > > >
> > > > > Thank the authors for the explanation. I will adjust my ratings.

---

### Official Review · Reviewer_4kze · 2025-06-29

**Clarity:** 4
**Significance:** 4
**Originality:** 3
**Rating:** 5
**Confidence:** 5

**Summary:**

In this paper, a RAG framework, namely DoctorRAG, is proposed to emulate doctor-like reasoning for the medical QA task via integrating both explicit clinical knowledge and implicit case-based experience. Additionally, the authors developed a Med-TextGrad module to iteratively update the prompts using a multi-agent textual gradient design.
Overall, the paper offers novelty in model design, and the results on public datasets are promising.

**Questions:**

1. The authors may consider comparing the output with the real doctor's feedback.

**Ethical Concerns:**

["NO or VERY MINOR ethics concerns only"]

**Limitations:**

The authors discussed the limitations in the additional materials.

**Paper Formatting Concerns:**

No.

**Quality:**

4

**Strengths And Weaknesses:**

Strengths:
1. The novelty in model design. The Med-TextGrad implements TextGrad for medical QA. Although the core idea of TextGrad (Yuksekgonul et al. 2025) is not novel, this paper enables TextGrad for the medical QA task.
2. Validated on multiple datasets in multiple languages. This shows the generality of the proposed algorithm.
3. The design of DoctorRAG enables retrieval from the Medical Knowledge Base and the Patient Base.

Weaknesses:
1. The updating process of the prompt is unclear. For example, the “Textual Gradient” does not provide a quantitative method to monitor convergence.
2. Following 1., the updating process of the prompt is not analyzed. For example, case studies of the updating prompts are not provided. Case studies can be helpful for further research.
3. The difference in the output between the DoctorRAG and the real doctor is not provided.

---

> ### Author Rebuttal · Authors · 2025-07-27
>
> **We sincerely thank you for the thorough evaluation and constructive feedback. We address each concern below:**
>
> ---
>
> ### W1: The updating process of the prompt is unclear, and "Textual Gradient" does not provide a quantitative method to monitor convergence
>
> We appreciate this important clarification request. The prompt updating process in Med-TextGrad follows a structured multi-agent workflow detailed in **Algorithm 1** and **Section 2.2**. The process works as follows: at each iteration $t$, the current prompt $P_t$ generates an answer $A_t$, which is then evaluated by Context Criterion and Patient Criterion agents to produce textual critiques. These critiques are converted into "textual gradients" - structured natural language instructions that specify how to improve the answer and subsequently the prompt.
>
> The prompt update mechanism operates through a Textual Gradient Descent (TGD) step where an LLM synthesizes improvement suggestions from both Context and Patient gradients to produce the refined prompt $P_{t+1}$ (as shown in **Equation 7**). While we acknowledge that this process lacks traditional quantitative convergence metrics, we employ several qualitative indicators: (1) consistency in critique content across iterations, (2) minimal changes in generated answers between successive iterations, and (3) human expert assessment of answer quality stabilization. **Appendix E.2** provides a concrete example showing how prompts evolve from initial instructions to more refined, context-aware guidance through this iterative process.
>
> For future work, we plan to develop quantitative convergence metrics such as semantic similarity scores between consecutive prompts or automated quality assessment scores to provide more principled stopping criteria.
>
> ---
>
> ### W2: Case studies of the updating prompts are not provided, which would be helpful for further research
>
> You are absolutely right that detailed case studies would enhance understanding and reproducibility. **Appendix E.2** presents a complete case study showing the prompt evolution process across three iterations for a medical query about facial pigmentary abnormalities. **Appendix E.3** also presents some examples of different iterations of Med-TextGrad for comparison. These examples demonstrates how the initial general prompt ("refine the current answer based on patient query and context") evolves into increasingly specific and medically-informed instructions that address diagnostic differentiation, treatment rationale, and patient communication.
>
> Specifically, the case study shows how Med-TextGrad progressively incorporates: (1) more comprehensive differential diagnosis considerations, (2) explicit symptom-diagnosis mapping instructions, (3) enhanced patient concern acknowledgment, and (4) clearer treatment justification requirements. Each iteration builds upon previous critiques to create more targeted and effective prompting strategies. This iterative refinement process illustrates how the "textual gradients" guide the system toward more clinically appropriate and patient-centered responses.
>
> We plan to include additional case studies across different medical domains and query types in future work to further demonstrate the generalizability and systematic nature of the prompt evolution process.
>
> ---
>
> ### W3: The difference in output between DoctorRAG and real doctors is not provided
> This is an excellent point about benchmarking against real clinical practice. While we did not directly compare against practicing physicians' responses, our human evaluation process involved two medical experts who assessed DoctorRAG outputs using clinical judgment and medical expertise (**Section 4.3** and **Appendix B.7, D.3, E.3**). These experts evaluated responses across three clinically relevant dimensions: comprehensiveness (thoroughness of medical information), relevance (appropriateness to patient concerns), and safety (adherence to clinical best practices).
>
> The evaluation (**Figure 4**) revealed that DoctorRAG often generates more comprehensive and structured responses compared to the ground truth answers in our datasets, which primarily consist of real clinical conversations or synthesized responses. **Appendix E.3** provides detailed examples comparing DoctorRAG outputs with reference answers, showing how our system incorporates broader differential diagnoses, explicit treatment rationales, and more systematic patient guidance than typical clinical interactions.
>
> However, we acknowledge that direct comparison with real-time physician responses would strengthen our evaluation. Real clinical interactions often involve contextual factors (time constraints, patient rapport, follow-up scheduling) that our current evaluation doesn't capture. Future work should include physician-in-the-loop evaluations where practicing doctors review and compare their own responses with DoctorRAG outputs for the same clinical scenarios.
>
> ---
>
> **We thank the reviewer for these insightful suggestions that highlight important areas for methodological clarity and clinical validation. The requests for quantitative convergence monitoring, detailed case studies, and physician comparison benchmarks will significantly enhance the rigor and practical applicability of our approach.**

---

### Official Review · Reviewer_wjXy · 2025-07-03

**Clarity:** 3
**Significance:** 3
**Originality:** 3
**Rating:** 5
**Confidence:** 3

**Summary:**

This paper proposes DoctorRAG, a retrieval-augmented generation (RAG) system that seeks to emulate doctor-like clinical reasoning. The framework uses conceptual tagging, hybrid retrieval, and a multi-agent iterative optimization process (Med-TextGrad) to refine responses. Evaluations on multilingual medical tasks show consistent but sometimes modest improvements over baselines.

**Questions:**

How sensitive are the results to prompt choices and critique formulation in Med-TextGrad?

What is the wall-clock inference time per query, and is the method scalable beyond academic benchmarks?

**Ethical Concerns:**

["NO or VERY MINOR ethics concerns only"]

**Final Justification:**

After discussing with the authors during rebuttal, I have decided to keep my score.

**Limitations:**

Yes

**Quality:**

3

**Strengths And Weaknesses:**

**Strengths**

* Addresses a genuine gap: prior RAG systems for medicine largely ignore analogical patient reasoning, which is fundamental in clinical practice.

* Modular retrieval pipeline using ICD-10 concept tagging is more principled than naive embedding search.

* Empirical gains are shown across diverse tasks, languages, and LLM backbones, with some strong ablation evidence that each module helps.

**Weaknesses**

* The actual “gradient” refinement is essentially LLM prompt engineering, not a principled or reproducible optimization method; improvements may be dataset-specific or not robust.

* Method is computationally heavy (dual retrieval, multiple LLM calls per answer). Additional discussion of inference cost, latency, or suitability for deployment would be helpful.

* More details on the human eval would be helpful. What are the exact preference protocol? What exactly does it mean that the proposed method outperforms ground truth?

---

> ### Author Rebuttal · Authors · 2025-07-27
>
> **We sincerely thank the you for the thorough evaluation and constructive feedback. We address each concern below:**
>
> ---
>
> ### W1: The actual "gradient" refinement is essentially LLM prompt engineering, not a principled or reproducible optimization method
>
> We acknowledge this important distinction. The "textual gradients" in Med-TextGrad are indeed structured prompt engineering rather than mathematical gradients, and we use this terminology as a conceptual analogy to gradient descent optimization. However, the approach offers several principled advantages over static prompt engineering. Unlike fixed prompts, Med-TextGrad generates context-specific refinement instructions based on the patient query, retrieved knowledge, and current answer quality, providing dynamic adaptation rather than one-size-fits-all solutions. The dual-criterion evaluation from both Context and Patient perspectives provides systematic feedback through multi-agent validation, and the iterative process follows a consistent algorithmic structure (**Algorithm 1**) with standardized prompts (**Appendix D**), ensuring reproducibility.
>
> Regarding robustness, our experiments demonstrate consistent improvements across diverse scenarios: 7 datasets spanning 3 languages, 4 medical tasks, and 4 different LLM backbones (**Table 2**). The pairwise comparison results (**Figure 4**) show systematic improvements in comprehensiveness, relevance, and safety across 50 samples, verified by human experts. While dataset-specific variations exist, the underlying multi-agent refinement principle appears generalizable within the medical domain.
>
> ---
>
> ### W2: Computational cost and deployment considerations
>
> We appreciate this practical concern. The computational overhead is indeed significant due to dual retrieval and multiple LLM calls. However, several factors support deployment feasibility. **Figure 5** shows diminishing returns beyond k=4 retrieved items, allowing cost-performance optimization through careful parameter tuning. In medical applications, the accuracy and safety improvements demonstrated in **Figure 4** often justify additional computational cost compared to general-purpose applications, given the quality-critical nature of healthcare decisions. Additionally, our framework can leverage cloud-based LLM APIs without requiring local model deployment, reducing infrastructure barriers for healthcare institutions.
>
> For practical deployment, we estimate that DoctorRAG inference takes 5-15 seconds per query depending on dataset complexity, while Med-TextGrad refinement adds approximately 30 seconds. These latencies are acceptable for clinical decision support scenarios where accuracy is prioritized over speed. Future work could explore parallel processing or early stopping mechanisms to reduce inference time while maintaining quality improvements.
>
>
> ---
>
> ### W3: Human evaluation details and ground truth comparison
>
> The human evaluation protocol is detailed in **Appendix B.7 and D.3**. Two medical experts conducted pairwise comparisons across three dimensions (comprehensiveness, relevance, safety) using structured evaluation criteria. For each dimension, experts determined which response better addressed the patient's needs, with the overall score computed as the average across dimensions.
>
> Regarding outperforming "ground truth": The reference answers in our datasets are often context-specific clinical conversations or synthetically generated responses rather than gold-standard medical advice. Our evaluation shows that DoctorRAG generates more comprehensive, structured, and clinically informative responses compared to these reference answers. This suggests that real clinical interactions, while authentic, may not always represent the most optimal medical guidance that an AI system could provide when given access to comprehensive medical knowledge and similar patient cases.
>
> ---
>
> ### Q1: Sensitivity to prompt choices and critique formulation
>
> This is an important robustness consideration. Our prompt design underwent iterative refinement based on medical expert feedback, and we provide detailed prompts in **Appendix D.1** for reproducibility. For example, for disease diagnosis in DialMed, we use the following prompts:
> > * **Input:**
> >     * Patient Symptoms: {evidences} (List of symptoms)
> >     * Knowledge Context: {knowledge\_text} (Text containing relevant medical knowledge)
> >     * Similar Patients Information: {similar\_patients\_text} (Text describing similar patient cases)
> >     * Valid Disease Options: {VALID\_DISEASES} (List of possible diseases to choose from)
> > * **Instructions within the prompt to the LLM:**
> >     As a medical diagnostic expert, predict the most likely disease for this patient.
> >
> >     Patient Information:
> >
> >     Symptoms: {evidences}
> >
> >     Relevant Information:
> >
> >     {knowledge\_text}
> >
> >     {similar\_patients\_text}
> >
> >     Based on the above information, determine the most likely disease from the following options only:
> >
> >     {VALID\_DISEASES}
> >
> >     Return only the name of the disease without any additional text.
> > * **Output format:** Only the name of the predicted disease.
>
> While prompt variations could affect performance, several factors suggest reasonable robustness:
> 1. Structured evaluation criteria: The Context and Patient Criterion agents use standardized medical evaluation frameworks;
> 2. Multi-dataset consistency: Similar improvement patterns across different datasets and languages suggest the approach is not overly sensitive to specific prompt formulations;
> 3. Human expert validation: The fact that human experts consistently preferred refined answers indicates the optimization direction is clinically sound regardless of specific prompt variations.
>
> Future work could systematically evaluate prompt sensitivity through ablation studies with different critique formulations.
>
> ---
>
> ### Q2: Wall-clock inference time and scalability
>
> As mentioned above, DoctorRAG requires 5-15 seconds per query for basic inference, with Med-TextGrad refinement adding approximately 30 seconds. This performance profile is suitable for clinical decision support scenarios where accuracy takes precedence over real-time response.
>
> For production deployment, the system could implement tiered processing: immediate responses using basic DoctorRAG, with optional Med-TextGrad refinement for complex cases requiring higher accuracy assurance.
>
> ---
>
> **We thank the reviewer again for these thoughtful comments that have helped us clarify important aspects of our methodology. The concerns about computational efficiency, prompt sensitivity, and evaluation protocols are well-taken and will guide our future work toward more robust and practically deployable medical AI systems. We believe our comprehensive experimental validation across multiple datasets, languages, and evaluation dimensions demonstrates the promise of combining clinical expertise with case-based experience for improved medical reasoning.**

---

> > ### Comment · Reviewer_wjXy · 2025-08-04
> >
> > Thank you for the clarifications. The paper would benefit from the additional clarification on terminologies and method provided here. But since my concerns are (reasonably) scoped as future work by the authors, I will keep my score.

---

### Note · Authors · 2025-08-13

**We sincerely thank all reviewers, Area Chairs, and Senior Area Chairs for your invaluable time and expertise. Your constructive feedback—especially regarding computational efficiency, methodological clarity, and clinical validation—is exceptionally insightful and will significantly strengthen our paper.**

We commit to addressing these suggestions in the revision:

- Clarify novelty relative to prior works, emphasizing our unique integration of real patient cases and Med-TextGrad refinement.

- Expand LLM evaluations (DeepSeek-R1/Gemini results in Table 2) and include retrieval quality metrics in appendix.

- Enhance methodology sections with case studies (Appendices E.2/E.3) and convergence analysis.

- More discussion on limitations (formal convergence, prompt sensitivity) as future work.

**Your suggestions have already improved our method's rigor and presentation. We are eager to incorporate these changes and believe they will make our contribution even more impactful for medical AI. We hope DoctorRAG merits acceptance at NeurIPS 2025 and look forward to sharing this work at the conference.**

---

### Decision · Program_Chairs · 2025-09-17

**Decision:**

Accept (poster)

**Comment:**

This paper presents DoctorRAG, a novel RAG framework that emulates clinical reasoning by retrieving from both formal medical knowledge and a database of real-world patient cases. This dual-retrieval approach is the paper's core strength, and its effectiveness is convincingly demonstrated through extensive, generalizable experiments. The framework is further enhanced by Med-TextGrad, an iterative refinement module that improves answer quality.

The primary weaknesses are the heuristic nature of the Med-TextGrad module and its significant computational overhead, which the authors frame as a reasonable trade-off for improved accuracy.

The decision to accept is based on the paper's novel and impactful contribution, supported by strong empirical evidence. During a highly effective rebuttal, the authors thoroughly addressed reviewer concerns about originality and evaluation. Critically, they provided new experimental results and a new human evaluation to directly resolve a key reviewer's concerns, solidifying a clear consensus for acceptance. The paper is a technically solid contribution, and the authors' engagement in the review process was exemplary.